# Zygote morphogenesis but not the establishment of cell polarity in *Plasmodium berghei* is controlled by the small GTPase, RAB11A

**Harshal Patil**[ᵒ], **Katie R. Hughes**[ID][ᵒ], **Leandro Lemgruber**[ID], **Nisha Philip**,
**Nicholas Dickens**[ID], **G. Lucas Starnes**, **Andrew. P. Waters**[ID] *

Wellcome Centre for Integrative Parasitology, University of Glasgow, Glasgow, Scotland, United Kingdom

ᵒ These authors contributed equally to this work.
* Andy.Waters@glasgow.ac.uk

**Data Availability Statement:** Data will be held within a public repository ENA/SRA under accession number PRJEB34537.

## Abstract

*Plasmodium* species are apicomplexan parasites whose zoites are polarized cells with a marked apical organisation where the organelles associated with host cell invasion and colonization reside. *Plasmodium* gametes mate in the mosquito midgut to form the spherical and presumed apolar zygote that morphs during the following 24 hours into a polarized, elongated and motile zoite form, the ookinete. Endocytosis-mediated protein transport is generally necessary for the establishment and maintenance of polarity in epithelial cells and neurons, and the small GTPase RAB11A is an important regulator of protein transport via recycling endosomes. PbRAB11A is essential in blood stage asexual of *Plasmodium*. Therefore, a promoter swap strategy was employed to down-regulate PbRAB11A expression in gametocytes and zygotes of the rodent malaria parasite, *Plasmodium berghei* which demonstrated the essential role of RAB11A in ookinete development. The approach revealed that lack of PbRAB11A had no effect on gamete production and fertility rates however, the zygote to ookinete transition was almost totally inhibited and transmission through the mosquito was prevented. Lack of PbRAB11A did not prevent meiosis and mitosis, nor the establishment of polarity as indicated by the correct formation and positioning of the Inner Membrane Complex (IMC) and apical complex. However, morphological maturation was prevented and parasites remained spherical and immotile and furthermore, they were impaired in the secretion and distribution of microneme cargo. The data are consistent with the previously proposed model of RAB11A endosome mediated delivery of plasma membrane in *Toxoplasma gondii* if not its role in IMC formation and implicate it in microneme function.

## Author summary

According to the WHO there was estimated to be over 200 million cases of malaria in 2017 and nearly half a million deaths. The disease is caused by specific species of

**Funding:** APW received support from The Wellcome Trust (Wellcome.ac.uk) Grant refs: 083811/Z/07/Z; 107046/Z/15/Z, WT104111AIA. HP received support from the European Commission (https://cordis.europa.eu/en) Grant name & ref: Evimalar 242095. The funders had no role in study design, data collection and analysis, decision to publish, or preparation of the manuscript.

**Competing interests:** The authors have declared that no competing interests exist.

*Plasmodium* which are passed between human hosts by a mosquito vector. In order to transmit through the mosquito the single-celled parasite undergoes many developmental changes as it morphs from non-motile blood forms to become a polarised and motile ookinete in the mosquito midgut. Transport of proteins within the cell during these critical morphological transitions relies on specific endosome vesicles to correctly target proteins within the parasite. We investigated the role of the RAB11A protein which is known to be involved in endosomal vesicle targeting to generate cellular polarity in other organisms. Because RAB11A is also essential for parasite growth in the mammalian host we used a promoter swap system to specifically switch off RAB11A in the sexual transmission stages. In the absence of RAB11A parasites were unable to form elongated, motile ookinetes and were unable to pass through the mosquito. Interestingly the parasites were able to form some of the (polarising) structures specific to ookinetes however full morphological transformation did not occur and the parasites were not motile. We show that although proteins are still delivered to the parasite surface, secretion is impaired and that the mutant parasites are smaller despite obvious microtubule formation implying that there is a deficit in delivery of membrane to the surface.

## Introduction

All zooites formed by apicomplexan parasites are polarized cells exhibiting an asymmetric distribution of content with defined apical distribution of the organelles (rhoptries and micronemes) involved in the recognition and colonisation of host cells. One of the early and critical events during mosquito transmission of malaria parasites (*Plasmodium* spp) is the initial development from (what is assumed to be) a non-polar zygote to a motile and polarized ookinete following fusion of a male and female gamete in the blood meal. This is essential for the parasite to be able to escape the hostile environment of the mosquito midgut and represents the greatest period of time that *Plasmodium* remains outwith a host cell. Although little is known about the generation of polarity in *Plasmodium* species, studies in other organisms have established that polarity is vital to carry out specific functions such as cell growth, migration, protein transport and invasion [1]. Indeed, all eukaryotes achieve cell polarity through a conserved set of proteins which includes signalling molecules of the *rho* family of GTPases, cytoskeleton assembly and recruitment, mobilization of proteins from the intracellular pool to the tip of growth via vesicle delivery [1]. Although, the *rho* family of GTPases such as Cdc42 and its homologues seem to be conserved in majority of organisms studied, there are species specific varieties of polarity determining proteins with little discernible general conservation [2]. The development and establishment of polarity can be regulated by internal factors such as protein trafficking, microtubules and actin dynamics [reviewed in [3]] and utilise both the exocytic and endocytic pathways to develop polarity [4]. The members of Ras GTPase subfamily are central to cell growth, differentiation and survival [reviewed in [5, 6]] and RAB proteins of which there are 11 in *Plasmodium* [7] belong to a small Ras GTPase family and regulate vesicle transport in eukaryotes.

RAB11A is involved in regulating vesicular traffic during the recycling of endosomes [8] and may assist in cytokinesis [9]. RAB11A is prenylated and interacts with phosphatidylinositol-4 kinases (PI4Ks) [10, 11] and its effectors such as RAB1-family of interacting proteins (FIPs) [12]. RAB11A has been shown to be essential in all apicomplexans tested including *Plasmodium berghei* [9]. RAB11A is expressed in all *Plasmodium* asexual blood stages with punctate protein localization in schizonts [7, 11] that by analogy with *T. gondii* may be an

association with rhoptries [13] [9]. *T. gondii* expressing a ddFKBP-Rab11A dominant negative form (ddFKBP-Rab11A$_{DN}$) shows reduced invasion of host cells by ~85%, consistent with a role in transport and release of rhoptry content during invasion [14]. Furthermore, preceding cell division, co-localization studies of GAP45 and RAB11A support the notion that RAB11A also mediates the delivery of GAP45 to the Inner Membrane Complex (IMC) and interacts with elements of the motility apparatus Myosin A Tail domain Interacting Protein/Myosin Light chain (MTIP/MLC) but plays no role in the organelle biogenesis and localization of sub-pellicular microtubules [9]. However, in *T. gondii*, the correct delivery of the major surface antigen SAG1 to the plasma membrane requires functional RAB11A as the ddFKBP-Rab11A$_{DN}$ line showed patchy (abnormal) instead of smooth (normal) localization of SAG1 [9]. RAB11A controls the necessary step after biogenesis of secretary organelles but before assembly of the motor complex during cell division [9]. These data suggest that RAB11A along with unconventional myosin controls the IMC assembly and budding of daughter cells. *Plasmodium* iterates the formation of zoites through its life cycle with their attendant features of an IMC and apical organelle formation raising the prospect that the roles of RAB11A might be also be reiterated. Indeed, the transcript encoding PbRAB11A is one of 370 transcripts which are stabilized and translationally repressed by the DOZI translational repression complex in gametocytes of the rodent infectious malaria model, *Plasmodium berghei* [15, 16], indicating its potential role in ookinete development. Therefore, further investigation of PbRAB11A function in the fertilised female gamete (zygote) may reveal insights into the development of polarity in extracellular ookinetes, in *P. berghei* and avoid the possible influence of its habitual intracellular niche.

Here, using a promoter swap strategy we show that PbRAB11A is essential for the transmission of *Plasmodium* through mosquitoes. Lack of PbRAB11A in ookinetes does not affect the formation of internal, polarised landmarks such as the apical complex and IMC in developing zygotes: meiosis and translation of stored mRNAs is also achieved. However, chromosomes remain decondensed 24 hours post fertilization and zygotes are morphologically arrested and remain spherical. These data suggested that PbRAB11A has no role in the establishment or development of zygote polarity in *P. berghei* and instead is necessary for morphological progression in a manner consistent with the model of PbRAB11A mediated delivery of plasma membrane proposed for daughter cell formation in *T. gondii* tachyzoites and *Plasmodium* asexual blood stage forms.

## Results

### PbRAB11A is expressed throughout *P. berghei* life cycle and localizes to the periphery and apical tip of the ookinete

The gene encoding *pbrab11A* (PBANKA_1418900) was transcribed in all stages of the *P. berghei* life cycle examined and was most abundant in gametocytes [17] (Fig 1A). A polyclonal antibody raised against the C-terminal end of PBRab11A (S4 Table) was used in Western analysis of the WT-GFP$_{CON}$, [18] line and demonstrated expression of *pbrab11a* across mixed asexual stages, schizonts, gametocytes and ookinetes. A smaller, ~23kDa band of unknown significance was detected indicating possible differential regulation or N-terminal processing of PbRAB11A in schizonts and mature ookinetes (asterisked in Fig 1B and 1C). Immunofluorescence microscopy using the same polyclonal antibody on fixed in WT-GFP$_{CON}$, parasites showed expression of PbRAB11A in mature schizonts (individual merozoites), 6h zygotes, ookinetes, oocysts as well as midgut sporozoites (Fig 1D). In merozoites, oocysts (day 10) and midgut sporozoites (day 10), the localization of PbRAB11A appeared to be cytoplasmic. In the 6h zygote, PbRAB11A was cytoplasmic and appeared punctate while in the ookinete PbRAB11A was more peripheral and with a focus at the apical tip suggesting a role of PbRAB11A at the apical complex (Fig 1D, S1 Fig).

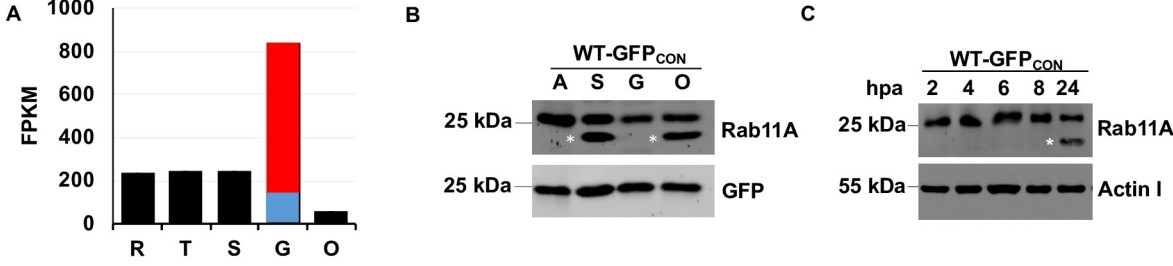

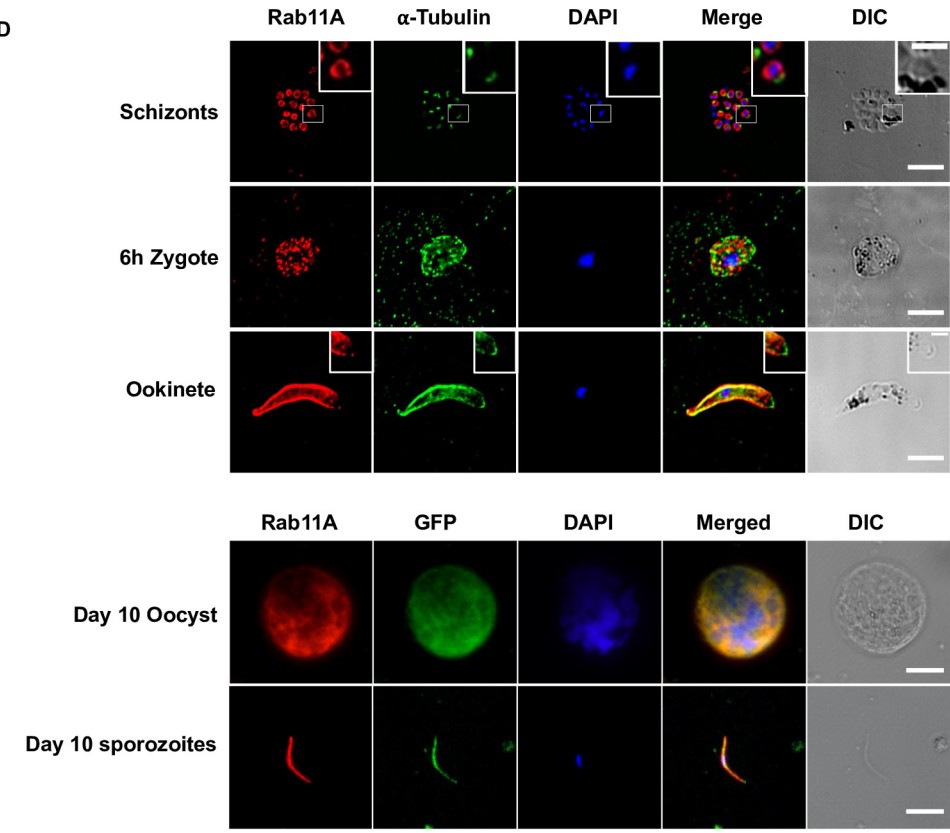

**Fig 1. Rab11A expression and localization during *P. berghei* life cycle stages. A.** PbRab11A transcription profile in FPKM at ring—R, trophozoite—T, schizont—S, gametocyte—G and ookinete—O stage [49]. Gametocyte expression data is shown as the measurements of Otto et al with the proportions of male (blue) to female (red) expression as reported by [50]. **B.** Expression of PbRab11A in: A, mixed asexual stages; S, schizonts; G, unactivated gametocytes and O, ookinete **C.** Expression of PbRab11A post-activation of gametocytes i.e. zygote to ookinete development using rabbit anti-PbRab11A antibody. Asterisks indicate the additional small PbRab11A band in schizont and ookinete. hpa = hours post-activation of gametocytes. **D.** Localization of Rab11A in WT-GFP_CON parasites determined through anti-PbRab11A antibody during blood stage: schizont (separated merozoites), mosquito stages: 6h zygote, ookinete, day 10 oocyst and midgut sporozoites. Insets show magnified images of a segmented merozoite (top) or an ookinete (bottom) tip in their respective panels. Scale bar 5 μm, Inset scale bar 2 μm.

## Rab11A is necessary to generate normal ookinete morphology, motility and for transmission

The essential nature of PbRAB11A in the asexual blood stages [9] required a conditional approach for gene disruption to study protein function in the ookinete stage. The promoter swap strategy that has been previously validated [17, 19, 20] is a reliable method to knock

down gene expression during the gametocyte to ookinete transition especially since modification of the PbRAB11A at the C-terminus would be expected to inactivate the protein. Both the *clag* (PBANKA_140060) and *ama-1* (PBANKA_091500) promoters are known to remain silent in sexual stages whilst maintaining sufficient expression levels during asexual blood stages [17, 19, 20]. 0.9 kb 5' UpStream Region (USR) of *rab11a* was replaced by 2 kb 5'USR of *clag* (referred to as $_p$*clag*) or 1.7 kb 5'USR of *ama-1* (referred to as $_p$*ama-1*) and a *2cmyc* N-terminal tag using conventional double crossover homologous recombination in WT-GFP$_{CON}$ parasites. Mutants were successfully selected and cloned via serial dilution [21]. Both independently generated *pbrab11A* promoter-swap mutants: $_p$*clag:2cmyc::rab11a* (line G480) and $_p$*ama-1:2cmyc::rab11a* (line G481) (henceforth referred as $_p$*clag::rab11a* and $_p$*ama-1::rab11a*) demonstrated appropriate genomic integration of the introduced constructs (S2 Fig).

The two independently generated *pbrab11A* promoter-swap mutants: $_p$*clag::rab11a* and $_p$*ama-1::rab11a* showed no apparent growth defects during blood stages and gametocytogenesis (analysed by morphological comparison of gametocytes, exflagellation of male gametocytes, ratio of male to female gametocytes in parasites (S3 Fig). However, visual inspection of *in vitro* ookinete cultures of $_p$*clag::rab11a* and $_p$*ama-1::rab11a* parasites revealed a severely impaired zygote to ookinete development by up to 99% and 98% respectively (Fig 2A–2C). Zygotes of both promoter swap lines remained spherical and failed to develop the characteristic elongated morphology of a wild type ookinete although the mature ookinete surface marker p25 was expressed and distributed to the surface of both promoter swap lines (Fig 2A–2C). Western analysis of $_p$*clag::rab11a* and $_p$*ama-1::rab11a* showed normal expression of RAB11A in mixed asexual blood stages, however, PbRAB11A is down-regulated in $_p$*clag::rab11a* and $_p$*ama-1::rab11a* gametocytes and ookinetes (Fig 2D and 2E). The additional smaller form of PbRAB11A noted in schizonts and mature ookinetes was only detected by the polyclonal PbRAB11A antisera and not by the anti c-MYC antibody indicating that the protein confirming that the smaller isoform lacks the N-terminus of the larger form and therefore the c-MYC epitope that would be present in the promoter swap lines. In addition to the full length protein, a slightly smaller RAB11A product is consistently detected by the anti-c-MYC antibody in the $_p$*clag::rab11a* line. The reason for this is unclear but may represent a processing event that was not characterised further (Fig 2D). The $_p$*clag::rab11a* line was arbitrarily chosen for the majority of the further analyses and western analysis demonstrated that zygotes of this line failed to produce significant levels of PbRAB11A from 2-24hpa (hours post activation) (Fig 2B, 2D and 2E). Immunofluorescence studies supported the down-regulation of PbRAB11A in $_p$*clag:: rab11a* "ookinetes" (Fig 2B). The expression of the paralogous PbRAB11B in $_p$*clag::rab11a* was unaffected during zygote to ookinete development (S4 Fig). Activated-Unfertilized 24h WT-GFP$_{CON}$ Female Gametes (AUFG—treated prior to activation with 2-Deoxy-D-glucose, 2DG [22] showed expression of PbRAB11A and P25 (Fig 2B and 2E) indicating activation of translationally stored mRNAs occurred and was not dependent upon fertilisation (S4 Fig). An imaging flow cytometry (IFC) strategy (S5 Fig, S6 Fig) was developed to quantitate the ookinete conversion rate and confirmed the results seen by conventional microscopy (Fig 2F). Using IFC the level of the P25 on the parasite surface was also measured and compared to the degree of ookinete development (stages I-IV as defined by Janse et al, 1985 [23]). This showed that P25, although still strongly expressed was expressed at slightly lower levels in the promoter swap lines compared to the control line (Fig 2G). IFC was also used to measure the perimeter of the parasite as a measure of the surface area of parasites. Mature elongated wild type ookinetes as expected had an increased perimeter measurement and the predominantly circular parasites of the promoter swap lines retained a lower perimeter measurement similar to immature zygotes and were therefore, of smaller surface area (Fig 2H). The images of the parasites taken during IFC from the different groups confirmed the accuracy of the classifications (Fig

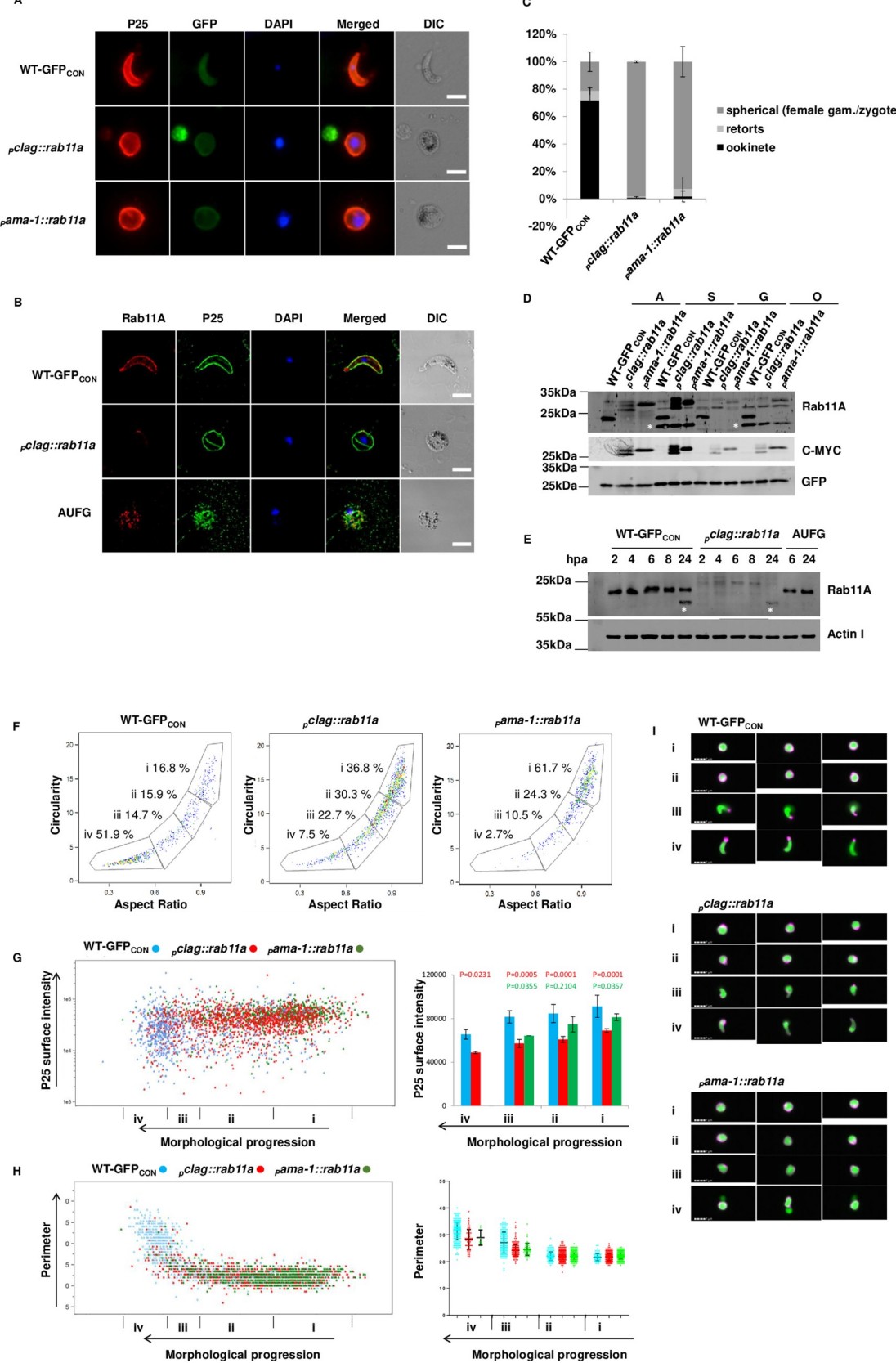

**Fig 2. Mutant _p_clag::rab11a and _p_ama-1::rab11a do not make ookinetes and down-regulate PbRab11A. A/B.** Microscopy images of immunofluorescence on fixed parasites using anti-RAB11A and/or anti-P25 antibodies as indicated and DAPI as a nuclear stain. **B.** Fertilized _p_clag::rab11a spherical ookinetes and **A**ctivated **U**nfertilised **F**emale **G**ametes (AUFG) i.e. 24h post-activation showing enlarged nucleus (Images taken on Axioplan). Scale bars 5 μm. **C.** Plot of _p_clag::rab11a and _p_ama-1::rab11a ookinete development (n = 3, mean +/-SD, two tailed student t test, p-value 0.0001). **D.** Western blot for Rab11A in _p_clag::rab11a and _p_ama-1::rab11a parasites at various life cycle stages probed with anti-PbRab11A, anti-c-MYC antibody. GFP is a loading control. Asterisks show the presence of extra bands in schizont and ookinete stages when the blot is probed with anti-PbRab11A antibody. A: Asexual mixed blood stage; S: Schizonts; G: enriched Gametocytes; O: ookinetes. **E.** Western blots showing the expression of Rab11A in zygotes and ookinetes produced by _p_clag::rab11a at 2, 4, 6, 8 and 24hpa as well as in AUFG 6 and 24hpa. **F-I.** Data from live parasites stained with anti-P25-Cy3 antibody, run on an Imagestream^x MkII imaging flow cytometer and analysed in IDEAS software. All figures show analysis of >750 activated cells after gating for activated parasites based on area focus, GFP and P25-Cy3 intensity (See S5 Fig) **F.** Imaging flow cytometry (IFC) density plots showing the aspect ratio of the masked GFP image (X axis) (AspectRatio_AdaptiveErode(M02, Ch02, 84)) against the circularity of the masked GFP image (Circularity_AdaptiveErode(M02, Ch02, 84)). Percentage values show proportion of parasites falling into each gate as described. **G.** Left panel: Dot plot of ookinete development (Circularity_AdaptiveErode(M02,Ch02,84)/AspectRatio(M02,Ch02,84) against P25 intensity (Intensity_MC_Ch03) for the WT-GFP_CON (blue) _p_clag::rab11a (red) and _p_ama1::rab11a (green). Right panel: Bar graph representation of data in left panel split into 4 gates as indicated below. Mean +/-SD and P value 2-tailed t test compared to the control line for each promoter swap line. **H.** Left panel: Dot plot of the ookinete development against a measurement of the periphery of the ookinete using the perimeter measurement feature in IDEAS software for WT-GFP_CON (blue) _p_clag::rab11a (red) and _p_ama1::rab11a (green). Right panel: Graphical representation of the data in the left panel for four gates as indicated below. Bars show median and interquartile range. **I.** Representative images from gates i-iv for each line showing overlay of channel 2 GFP (green) and channel 3 antiP25-Cy3 antibody (magenta).

2I). The _p_clag::rab11a zygotes were able to complete meiosis (S7 Fig) although apparently with reduced success compared to WT parasites although this was not a statistically significant difference. Both male and female *rab11a* alleles appeared to contribute to the full development of the ookinete since crosses with P47ko and P48/45ko lines which make competent male or female gametes respectively only partially restored the ability to generate wild type numbers of mature, morphologically normal ookinetes (S8 Fig). The nucleus of the _p_clag::rab11a "ookinetes" appeared decondensed suggesting a qualitative impact on DNA replication or organisation (Fig 2A & S8 Fig).

To determine if the mainly spherical _p_clag::rab11a and _p_ama-1::rab11a ookinete populations were able to complete the life cycle through their invertebrate host, mosquito transmission experiments were performed (Fig 3). Female *Anopheles stephensi* mosquitoes were allowed to feed on WT-GFP_CON, _p_clag::rab11a or _p_ama-1::rab11a infected mice. Mosquito midguts were examined for the presence of oocysts on day 11 or 14. Mosquito midguts were also analysed on day 17, 18 or 22 assuming that _p_clag::rab11a and _p_ama-1::rab11a spherical ookinetes might have delayed midgut transversal and oocyst development. The WT-GFP_CON showed normal oocyst development in mosquito midguts [median 150, mean 227 oocysts per midgut (n = 3)] while _p_clag::rab11a (maximum of 4 oocysts per midgut with small size, mean 0.269, median 0 and n = 3) (Fig 3A and 3B) or _p_ama-1::rab11a (maximum of 5 oocysts per midgut with small size, mean 0.31, median 0, n = 2) (independent WT-GFP_CON control for _p_ama-1::rab11a showed mean 84, median 130.47, n = 2) showed greatly reduced numbers of oocysts and salivary gland analysis showed a complete absence of _p_clag::rab11a and _p_ama-1::rab11a sporozoites (S9 Fig). Upon feeding of infected mosquitoes on mice (bite-back) on day 18, 22 or 24, no parasites were observed in _p_clag::rab11a (n = 3) or _p_ama-1::rab11a (n = 2) infected mice which were monitored until day 14 post infection while WT-GFP_CON parasites were observed on day 3 in all experiments (S1 Table and S2 Table). Therefore, PbRAB11A is essential for the transmission of *P. berghei* and its role in the generation of ookinete morphology appears critical.

Due to the inability of _p_clag::rab11a ookinetes to infect mosquito midguts and their spherical shape, the motility of _p_clag::rab11a spherical ookinetes was examined in a comparative assay. 24hpa WT-mCherry and GFP-positive _p_clag::rab11a zygotes (i.e. ookinetes) were

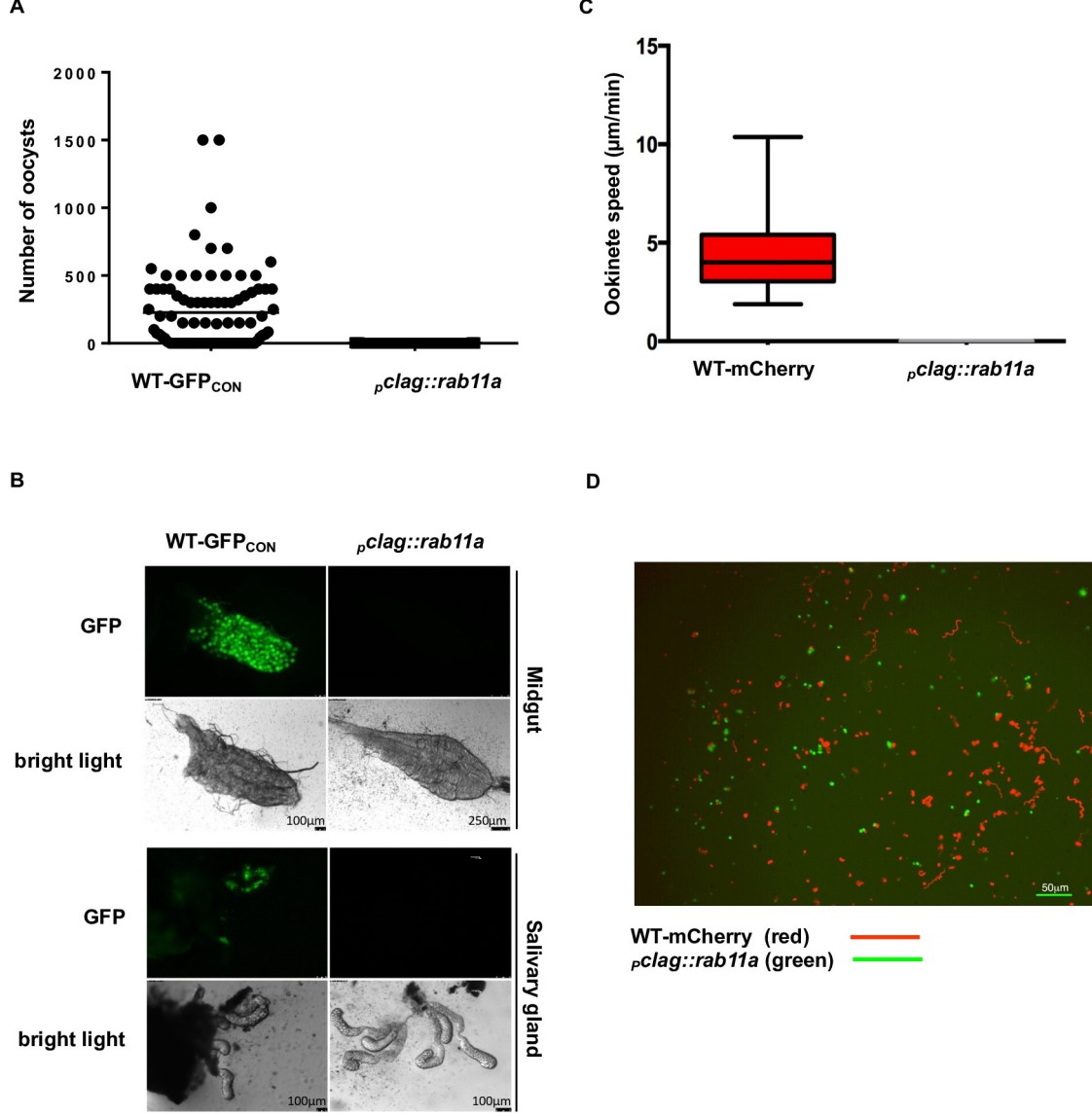

**Fig 3. _p_clag::rab11a parasites are unable to transmit through mosquitoes. A.** Plot of oocyst load on day 14 after feeding in dissected midguts of WT-GFP_{CON} or _p_clag::rab11a fed mosquitoes (n = 3, two tailed student t test, p-value 0.0001). **B.** Images of dissected midguts and salivary glands from WT-GFP_{CON} and _p_clag::rab11a parasites fed mosquitoes. **C.** Speed of WT-mCherry and _p_clag::rab11a spherical ookinetes. 16 ookinetes for each genotype were measured; bottom and top boxes denote first and third quartiles respectively, whiskers denote minimum and maximum, P<0.0001. **D.** Representative path of WT-mCherry and _p_clag::rab11a ookinetes for 10 minutes.

embedded in the same Matrigel preparation (see Materials and Methods) and their motility observed in the red and green fluorescent channels. WT-mCherry ookinetes possessed normal corkscrew like movement and an average speed of ~4 μm/min (range 2–10 μm/min) whereas _p_clag::rab11a spherical ookinetes were completely immobile (n = 2) (Fig 3C and 3D). This indicates that _p_clag::rab11a ookinetes are immobile either due to the spherical morphology caused by lack of PbRAB11A and/or possible lack of motility associated proteins and therefore, transmission through mosquitoes is blocked.

## PbRAB11A is not necessary for the generation and deposition of the apical complex and the IMC

The demonstration that PbRAB11A production is necessary for the generation of ookinete, morphology, motility and infectivity prompted a further examination of the mutant parasite ultrastructure using electron microscopy. Scanning electron microscopy (SEM) demonstrated that 8h WT-GFP$_{CON}$ zygotes have progressed to the retort form while 8h $_p$*clag::rab11a* zygotes have a small outgrowth of plasma membrane possibly indicating a site for apical complex development (yellow arrow, Fig 4A). Transmission electron microscopy (TEM) of the same samples demonstrated that the IMC had been fully developed as well as the apical complex in

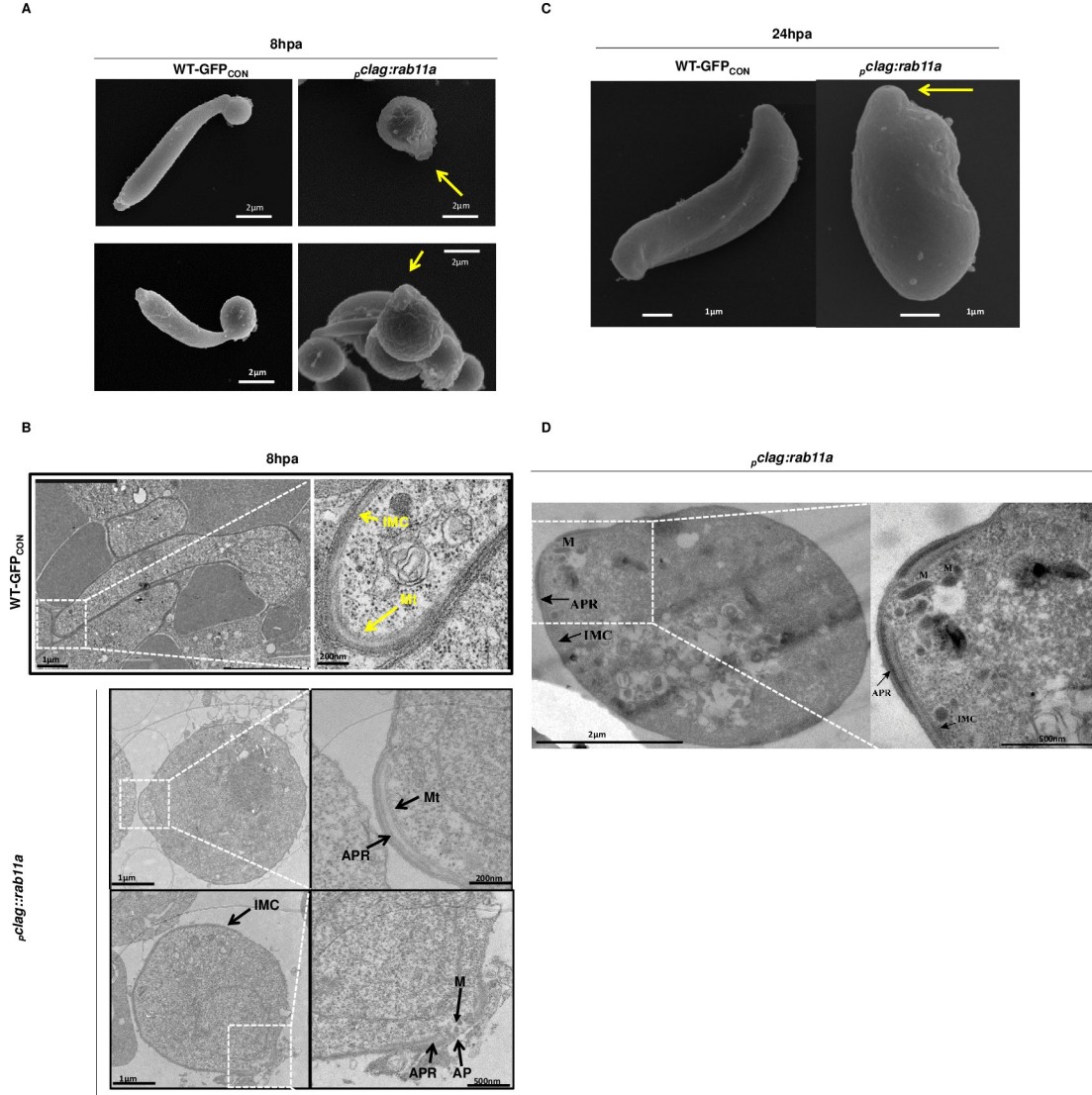

**Fig 4. Ultrastructural analysis of $_p$*clag::rab11a* 8h zygotes. A.** SEM images of 8h $_p$*clag::rab11a* zygotes showing the typical small yet specific membrane extension (see yellow arrow) as compared to retort outgrowth in 8h WT-GFP$_{CON}$ zygotes. **B.** TEM images of 8h $_p$*clag::rab11a* zygotes showing integrity of Apical Ring (APR) with aperture (AP), IMC, subpellicular microtubules (Mt) and micronemes (M). **C.** SEM of left, 24h WT-GFP$_{CON}$ and right, $_p$*clag::rab11a* "ookinetes", the apical prominence is arrowed in the latter. **D.** TEM images of two magnifications of 24hpa $_p$*clag::rab11a* "ookinetes" demonstrating the fully developed nature of the apical prominence.

8h ~p~*clag*::*rab11a* zygotes with readily identifiable Apical Ring (APR) with an aperture (AP), micronemes (M), Inner membrane complex (IMC) and subpellicular microtubules (Mt) (Fig 4B). EM imaging of parasites after 24 hours of development showed the development of the ~p~*clag*::*rab11a* zygotes has barely progressed whereas the WT-GFP~CON~ parasites exhibited the expected, fully elongated form of mature ookinete (Fig 4C and 4D). This suggested that a full (and functional) apical complex was developed by ~p~*clag*::*rab11a* zygotes consisting of the IMC and internal organelles (see below) somewhat in contrast to the role for RAB11A in the delivery of surface and IMC markers in *T*. *gondii* [9].

The expression and distribution of a number IMC and apical organelle components by developing ~p~*clag*::*rab11a* zygotes for which antibody reagents were available was, therefore, examined both by Western blot analysis and where possible immunofluorescence, comparing them to wild type parasites or activated unfertilised female gametes (AUFG) as controls. Western analysis of glideosome associated proteins: MTIP, GAP45 and GAP50, MyoA suggested a possible marginal delay in their expression between 6 to 8hpa and overall reduction in protein expression in ~p~*clag*::*rab11a* zygotes (Fig 5A). Similarly, the expression of ookinete secreted or surface distributed proteins (chitinase and P28) appeared to be delayed and/or reduced (Fig 5A). The expression of cytoplasmic and more general structural proteins seemed unaffected (DOZI, CITH and alpha-tubulin). The expression of a protein known to be associated with the apical tip of the developing ookinete, the protein phosphatase PPKL seems unaffected in the ~p~*clag*::*rab11a* line consistent with the observations that the apical complex is established as normal in the absence of PbRAB11A (Fig 5A, S10 Fig).

Immunofluorescence microscopy revealed that all IMC (GAP45, IMCb, MTIP, MyoA), invasive organelle (Chitinase, CTRP) and surface markers (P25) appear to be distributed where expected within the spherical mutant ~p~*clag*::*rab11a* matured zygotes (Fig 5, S10 Fig). In addition, PPKL serves as a marker for the apical tip of the mature ookinete and occupies a similar discrete location in ~p~*clag*::*rab11a* matured zygotes alongside evident microtubule formation (S10 Fig). Furthermore, the localised dynamics of IMC protein deposition seemed similar despite the morphological differences between the ookinetes. The localization of GAP45 in ~p~*clag*::*rab11a* zygotes was similar at 4h post-fertilization to WT-GFP~CON~, which indicates an initial focal point or bud development site from which the leading apical end will emerge (S11 Fig) However, although ~p~*clag*::*rab11a* 6h zygotes lack a prominent retort outgrowth, GAP45 is deposited in this mutant along the IMC showing an arc-like localisation and apparently bordering the whole IMC at 24h post fertilization (Fig 5B). The IMC is linked to the subpellicular microtubule network that is critical for parasite motility and morphology. To further investigate the development of this tubulin-based structure we used a live-cell permeant tubulin stain to perform IFC analysis on live parasites at different stages of development. Analysis of live parasites showed that activating male gametocytes and male gametes could be clearly visualised and the intensity and distribution of tubulin was unchanged in ~p~*clag*::*rab11a* compared to the control (Fig 5C and 5D). At 4 hpa activated females identified by their P25 surface marker showed a discrete focus of tubulin stain representing the formation of the apical complex and site for outgrowth during subsequent parasite development (Fig 5E). This point was clearly present in both WT and mutant parasites, again suggesting that Rab11A is not essential for defining polarity in developing ookinetes. By 24 hpa the tubulin stain was unable to permeate through the membrane of live ookinetes but following partial permeabilisation using L-α-Lysophosphatidylcholine (LysoPC) we were able to visualise the tubulin based cytoskeletal structure in some mature live ookinetes. The promoter swap mutant lines showed tubulin staining around the parasite (Fig 5F) again showing similar intensity and distribution around the periphery of the incompletely formed "ookinetes" compared to WT control ookinetes.

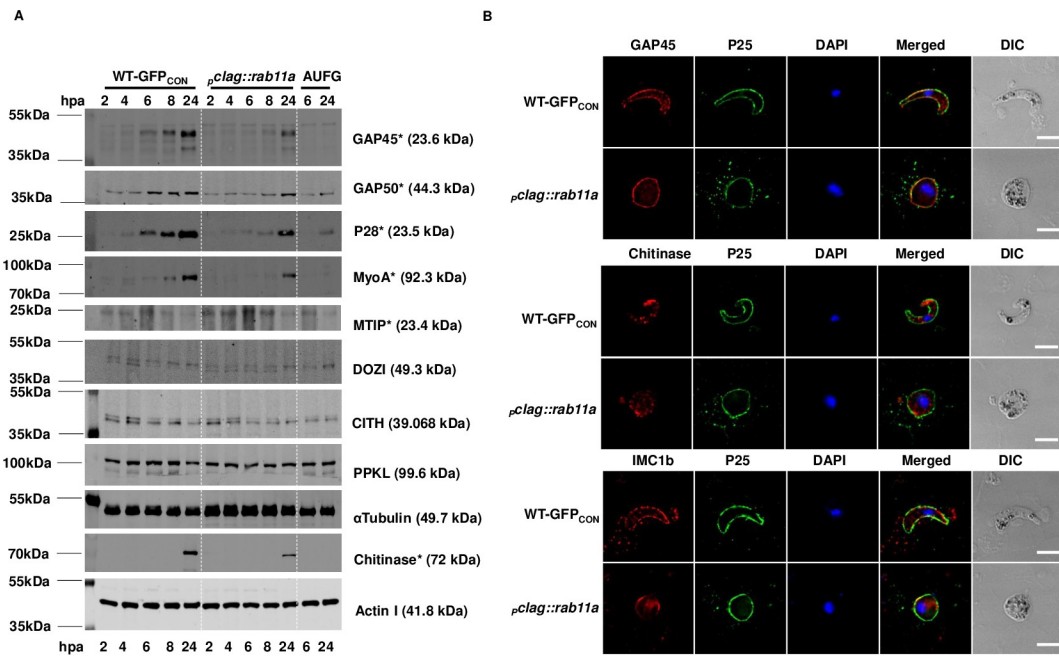

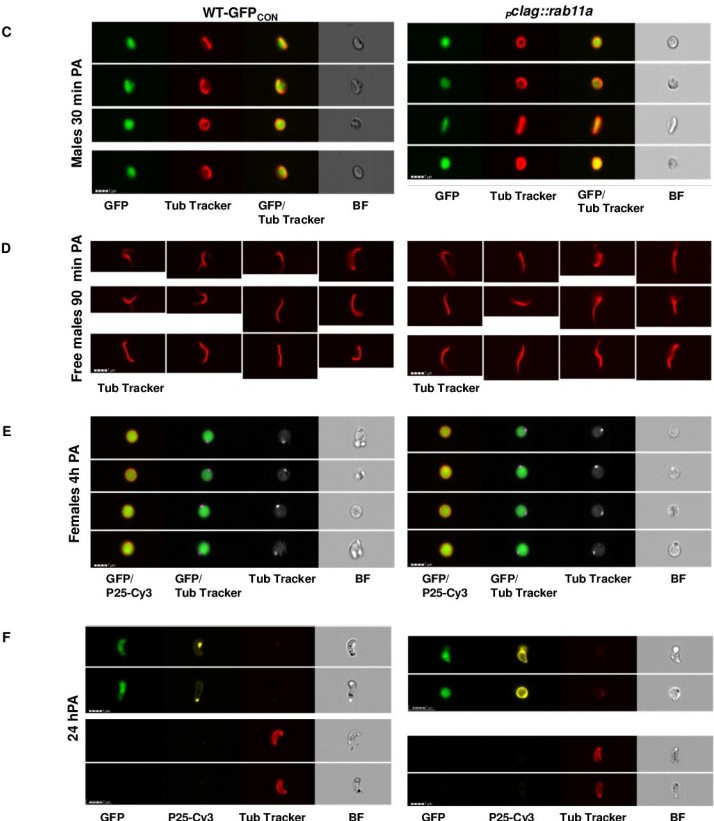

**Fig 5. Western and immunofluorescence microscopy of ookinete development and structural markers. A.** Western blots for WT-GFP_CON and *pclag::rab11a* gametocytes at 2, 4, 6, 8 and 24hpa, and for AUFG at 6 and 24hpa using anti-GAP45, anti-GAP50, anti-P28, anti-MyoA, anti-MTIP, anti-DOZI, anti-CITH, anti-PPKL, anti-αTubulin, anti-Chitinase and anti-actin I

antibodies. Actin I and α-Tubulin act as loading controls. Asterisks show down-regulated or delayed expression proteins in respective bands. **B.** Fixed WT-GFP$_{CON}$ ookinetes and $_pclag$::*rab11a* spherical ookinetes were probed with primary antibodies: anti-GAP45, anti-Chitinase or anti-IMC1b antibodies mixed with FITC-tagged anti-P25 antibody. Images shown are single slices of Deltavision deconvoluted Z stack. Scale bar 5 μm. **C-F.** Image galleries from IFC analysis. Samples were run on an Imagestream$^x$ MkII imaging flow cytometer were analysed using IDEAS software. Images are representative of over 200 gametocytes analysed. **C.** Live 30 min post activation (PA) parasites stained with anti-P25-Cy3 and Tubulin tracker (Invitrogen). Cells were gated for area, focus, GFP and absence of P25-Cy3 expression to identify male gametocytes. **D**. Live parasites 90 min PA gated for area, aspect ratio and tubulin tracker (Invitrogen) stain to identify free male gametocytes. **E.** Female gametocytes 4 hpa identified by gating for area, focus, GFP and P25-Cy3 expression. Overlaid images are shown for channel2 (GFP) and channel3 (P25-Cy3) and channel2 (GFP) and channel 5 (Tubulin tracker) and single channel images for Tubulin tracker and brightfield. Scale bar (lower left) = 7 μm. >200 female parasites were analysed and there was no difference in the proportion of parasites activating (P25-Cy3 positive) or displaying a prominent tubulin focus. **F.** Live parasites at 24 hpa stained with tubulin tracker showing non-permeabilised cells (upper) and after partial permeabilisation of cells with LysoPC (lower). Scale bar (for all images) = 7 μm.

In order to assess possible transcriptional consequences linked to the morphological defects, RNA-seq analysis was performed on both gametocytes and cells resulting from 24 hour cultures of activated gametocytes that would normally generate ookinetes using wild type and $_pclag$::*rab11a* parasites. Little transcriptional effect was noted in gametocytes (9 transcripts upregulated more than 2-fold and 49 similarly downregulated, S12 Fig) and there was no significant overlap of the de-regulated transcripts with the pool of transcripts that are destabilised in the absence of CITH or DOZI (1 transcript encoding a conserved protein of unknown function, PBANKA_0707700, S13 Fig). GO-term analysis did not reveal functional classes of transcript that were affected by the absence of PbRAB11A. There was a slightly stronger effect on ookinete transcription reflected in the number of transcripts with altered steady state level (32 downregulated, 84 upregulated, S14 Fig).

In the absence of obviously significant differences in protein expression, deposition and transcription, furthermore taking into account of the expected role of PbRAB11A in microneme biogenesis, we examined the functionality of secretion from the apical organelles which is known to be important for the progressive surface distribution of motility associated proteins (e.g. CTRP) and soluble enzymes associated with life cycle progression (e.g. chitinase) (reviewed in [23]). Using chitinase as the marker for secretion of apical organelle cargo we compared relative amounts produced by wild type and the two mutants $_pclag$::*rab11a* and $_pama1$::*rab11a* parasites after 24 hours of culture of activated gametocytes when mature ookinetes would normally be present. We compared levels of parasite-associated protein in gently pelleted parasites and amounts released into the culture supernatant using a soluble non-secreted cytoplasmic protein (DOZI) as a marker for parasite lysis and normalisation. The change in ratio of the amount of chitinase and DOZI in the supernatants indicated reduced production/secretion of chitinase was evident in both mutants (Fig 6).

## Discussion

Obligate intracellular protozoan parasites such as those represented by the Apicomplexa are likely to have further evolved the endocytic pathway common to all eukaryotes to suit the specialised needs of their intracellular niche. RAB11A is a highly conserved protein that can serve as a marker of both regulated and constitutive secretory pathways that may also ultimately deliver proteins to the cell surface [24]. *Plasmodium* spp. also have an extended period where they exist as an extracellular parasite in the blood meal of the mosquito following gamete formation during the development of the zygote to form the motile ookinete that will subsequently encyst at the basolaminar side of the mosquito midgut. This extracellular phase of the life cycle offers an opportunity to study the development of a stable, polarised apicomplexan cell without the influence of a surrounding host cell. Secretory activities (including those

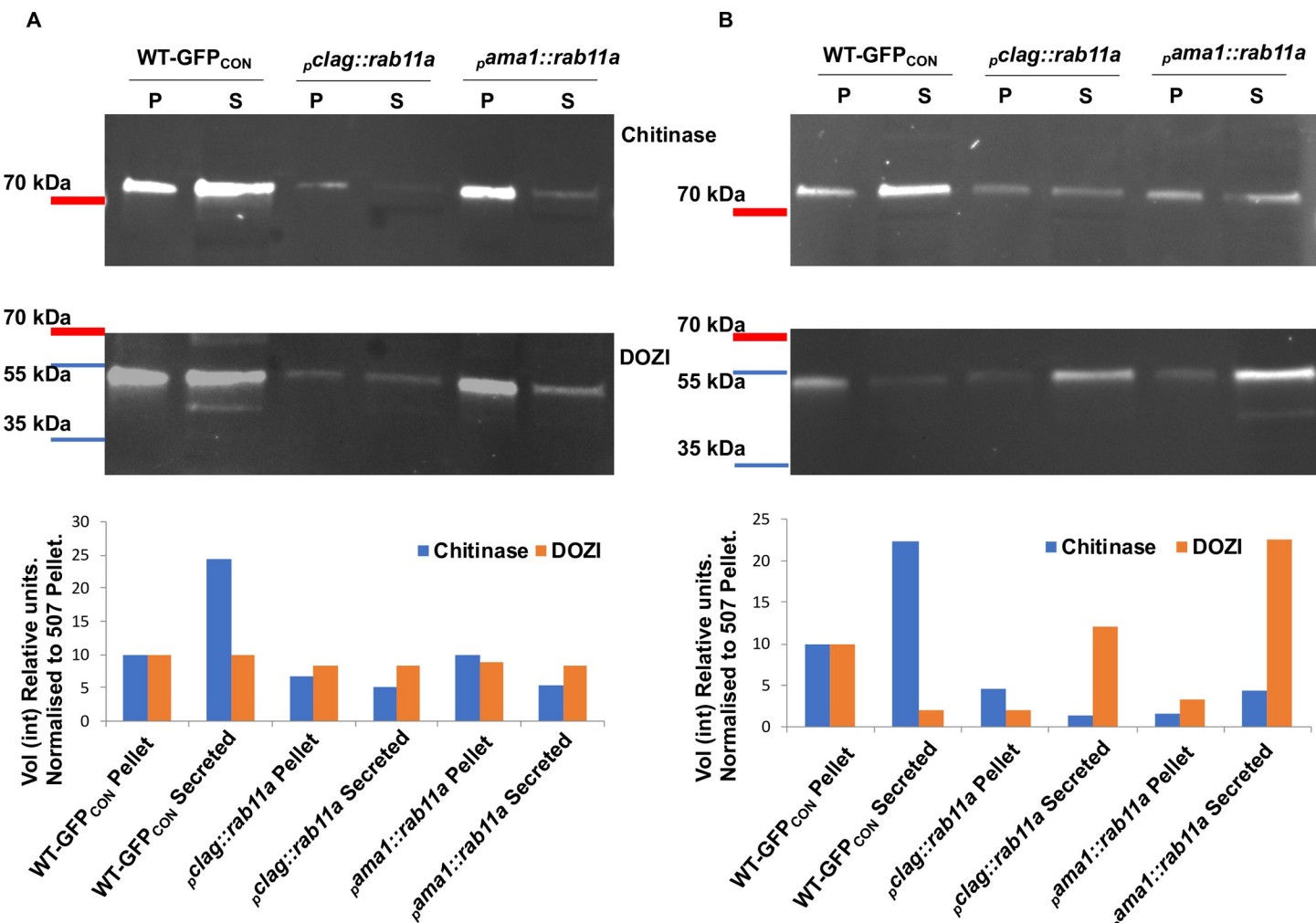

**Fig 6. Ookinetes deficient in PbRAB11A expression also are relatively deficient in protein secretion.** Representative Western blot analysis of culture parasite pellet and supernatants protein content analysing parasite pellets (P) after collection by centrifugation and in the remaining culture supernatant (S). Chitinase is expected to be released and shed into the culture supernatant respectively. DOZI is an abundant cytoplasmic protein that acts as a control for cell lysis. Two representative experiments are shown (A & B). Lower panels are quantitated according to the signals in the upper images and normalized relative to wt parasites.

involving PbRAB11A) will be confined to the delivery of membrane bound or secretory proteins to the parasite cell surface or to the specialised organelles associated with ookinete invasion (micronemes) for subsequent programmed release.

Previous attempts to delete PbRAB11A in haploid blood stages of *P. berghei* were unsuccessful indicating essentiality [9]. The abundance of *rab11A* mRNA in gametocytes [17] and the expectation that this mRNA was translationally repressed [15] suggested a role for PbRAB11A in ookinete development. We employed a c-MYC N-terminal tagging and a promoter swap strategy [19, 20] in order to avoid alteration of the C-terminus which contains a highly conserved and essential pair of cysteine residues necessary for the conserved geranylgeranylation required for RAB11A function [25];[26]. Two independent mutant parasite lines: *pclag::rab11a* (G480) and *pama-1:rab11a* (G481) exploiting the late schizont *clag* and *ama-1* promoters respectively, were shown to facilitate the blood stage asexual expression of PbRAB11A to normal levels and showed no obvious growth defect. However, *rab11a* was effectively silenced in the sexual stages allowing examination of protein function during

gametocyte and zygote development. Our results also extended the analysis of the expression of PbRAB11A during the *P. berghei* lifecycle (Figs 1 & 2) with Western analysis suggesting that two isoforms of PbRAB11A are produced in schizonts and ookinetes due to N-terminal processing. However, the significance of the observation is unclear and requires further investigation.

The absence of PbRAB11A in the $_p$*clag*::*rab11a* and $_p$*ama-1*::*rab11a* lines resulted in the formation of spherical ookinetes with an intact apical complex and IMC. Transcription and translation were also shown largely to be uncoupled to cellular development as both protein and transcriptome analyses revealed no striking differences from wild type gametocytes and ookinetes. Similarly spherical (and developmentally arrested) ookinetes of *P. berghei* were observed when *gap45* KD [27] and *pdeδ* KO [28] lines were analysed. However, the similarities are superficial given that GAP45 is expressed with only a slight delay and the IMC is correctly anchored in the $_p$*clag*::*rab11a* spherical ookinetes unlike in the *gap45* KD equivalents where the apical complex was observed to float freely inside the cytoplasm [27]. GAP45 is believed to be delivered to the IMC by Rab11A-mediated vesicles [9] and is also responsible for retort formation [27]. Localization studies of GAP45 suggest the selection of a focal point in $_p$*clag*::*rab11a* 4h zygotes which then disperses along the membrane of $_p$*clag*::*rab11a* 6h zygotes lacking the WT retort outgrowth and lays down the IMC across the entire cytoplasmic side of the plasma membrane of the spherical $_p$*clag*::*rab11a* ookinete (24hpa) indicating the assembly of IMC (S11 Fig and Fig 5B). Our data, however, suggest that delivery of GAP45 does not depend on PbRAB11A mediated vesicles at this stage of parasite development. Equally in ddFKBP-Rab11A$_{DN}$ tachyzoites expressing a dominant negative form of TgRAB11A the membrane surface protein, SAG1 was not delivered correctly to the cell surface [9] yet delivery of an equivalent surface marker in the developing *Plasmodium* zygote, P25, seems unaffected and smoothly decorates the surface.

SEM analysis revealed a small, cone-like outgrowth in 8h $_p$*clag*::*rab11a* zygotes (Fig 4A). Analysis of 8h $_p$*clag*::*rab11a* zygotes strongly suggest the outgrowth is associated with the assembly of the collar with an aperture, apical microtubules and the IMC (Fig 4A and 4B). Therefore, assembly of complete set of internal organelles such as IMC, apical complex and micronemes is expected in spherical $_p$*clag*::*rab11a* ookinetes and is somewhat comparable with the morphology of round dedifferentiated PDEδ KO *P. berghei* ookinetes which although they also form apical organelles, possess an incomplete IMC [28]. These round otherwise terminally differentiated PDEδ KO *P. berghei* ookinetes were shown to rotate rapidly but have little or no forward motility. The same motility assays suggest that $_p$*clag*::*rab11a* spherical ookinetes do not exhibit even rotational movement (Fig 3C and 3D) indicating a distinct developmental lesion. ImageStream analysis of $_p$*clag*::*rab11a* "ookinetes" indicated that these cells had a smaller surface area than the wild type ookinete equivalents and were therefore smaller. A recent study investigating delayed death phenotype in apicomplexan parasites has implicated a prenylation dependent role for RAB11A in the delivery of vesicular cargo (including GAP45) and intracellular trafficking [30]. Our data would be consistent with a deficit in delivery of lipid to the surface due to the absence of RAB11A implying there is also a stage specificity to the role of RAB11A and its cargo.

Sub-pellicular microtubule number varies greatly in the different zooite forms of *Plasmodium* (summarised in [29]) with ookinetes typically expressing around 60 [30] [31], merozoites 3–4 [32] and P. berghei sporozoites 16 [29]. The failure of our RAB11A mutant retorts to fully elongate might indicate that there was a physical uncoupling of the IMC from the sub-pellicular microtubules. Direct investigation of this using tubulin tracker in live zygotes/ookinetes and TEM analysis of fixed samples demonstrated an early event at 4hpa visualised as a single centre of polarised nucleation of tubulin subtending the plasma membrane. Unfortunately,

ookinetes become impermeable and live tracking of the microtubule development and organisation is not possible. Although TEM evidence was inconclusive it did demonstrate a regular array of microtubule origins around the apical prominence. Furthermore, immune-staining of tubulin indicated that microtubules are formed in parasites in the absence of RAB11A. Microtubule length can be a function of type and levels of expression [29], however, there was no apparent impact of lack of RAB11A on tubulin abundance. Despite the abundance of microtubules in the ookinete the function of RAB11A does not appear to involve microtubule formation nor tubulin expression.

Although no major perturbances in transcription profile were detected in the $_p$*clag*::*rab11a* gametocytes and ookinete forms, there was a minor temporal delay in protein expression of certain proteins assayed with our panel of antibodies. Furthermore, the expression of microneme proteins CTRP (also associated with ookinete motility) and chitinase appears to be reduced in $_p$*clag*::*rab11a* and $_p$*ama1*::*rab11a* ookinetes than WT-GFP$_{CON}$ ookinetes although micronemes are clearly formed (Fig 4D). Secretion assays confirmed the reduced expression of chitinase and demonstrated that there was no obvious secretion of chitinase indicating functional impairment of apical organelle secretion. A deficiency in secretion is also supported by the immunofluorescence data which indicate that both CTRP and chitinase localise in a pattern consistent with deposition in the apical organelles.

Based on current results and past studies, we propose a model for developmental block of $_p$*clag*::*rab11a* spherical ookinetes (Fig 7). The essential apical marker ISP1 is polarised in late female zygotes indicating pre-selection of the focal point of ookinete [33]. Published data [27] and our results (Fig 5, S11 Fig) suggest that GAP45 is also important for further development of the focal point, however, our data demonstrate that along with GAP45, PbRAB11A is critical for the retort outgrowth of *P. berghei* zygote. Once the focal point is marked through ISP1, GAP45 might start assembling at the focal point 4h post-fertilization along with as yet unknown focal point markers but which might include components of the IMC, collar and MTOC [34–36] through which microtubules extend. Subsequently, PbRAB11A along with cytoskeletal components such as actin as well as microtubules assist in the retort shape formation. Specifically, PbRAB11A endosomes might provide plasma membrane to the growing tip or to the joint of the retort and the main body of the zygote (here we refer to it as the 'neck' region) (Fig 7) and anticipate a role for Rab11A in the secretion, delivery of membrane synthesis components e.g. PI4K [11] and other necessary proteins while donating plasma membrane to assist the intended morphological transformation. Along with plasma membrane, the PbRAB11A endosome might help directly or indirectly in the incorporation of marker(s) at the neck region forming a transition point segregating the initial spherical body of the zygote from the retort outgrowth. Forces might be generated at the transition point that pushes the growing apical complex outwards by donating plasma membrane. Furthermore, the transition point 'neck marker/s' may act as a centre for pulling the plasma membrane during ookinete development. In $_p$*clag*::*rab11a* zygotes, all characterised developmental and structural markers are formed normally however, we hypothesise that PbRAB11A-mediated endosomes are responsible for delivering the additional plasma membrane required to form the ookinete as it is larger than the female gamete and early zygote. Normally such membrane structures would be both recycled and *de novo* synthesised and made available for the growing apical complex. In the absence of Rab11A this does not occur, neck markers are probably mis-localized however, apical bud formation is achieved but not further outgrowth.

The zygote developmental processes described here serve as models for the developmental events of intracellular forms of *Plasmodium* such as sporozoites and merozoites as well as other apicomplexans that might be exploited for disease interventions. Our data highlight,

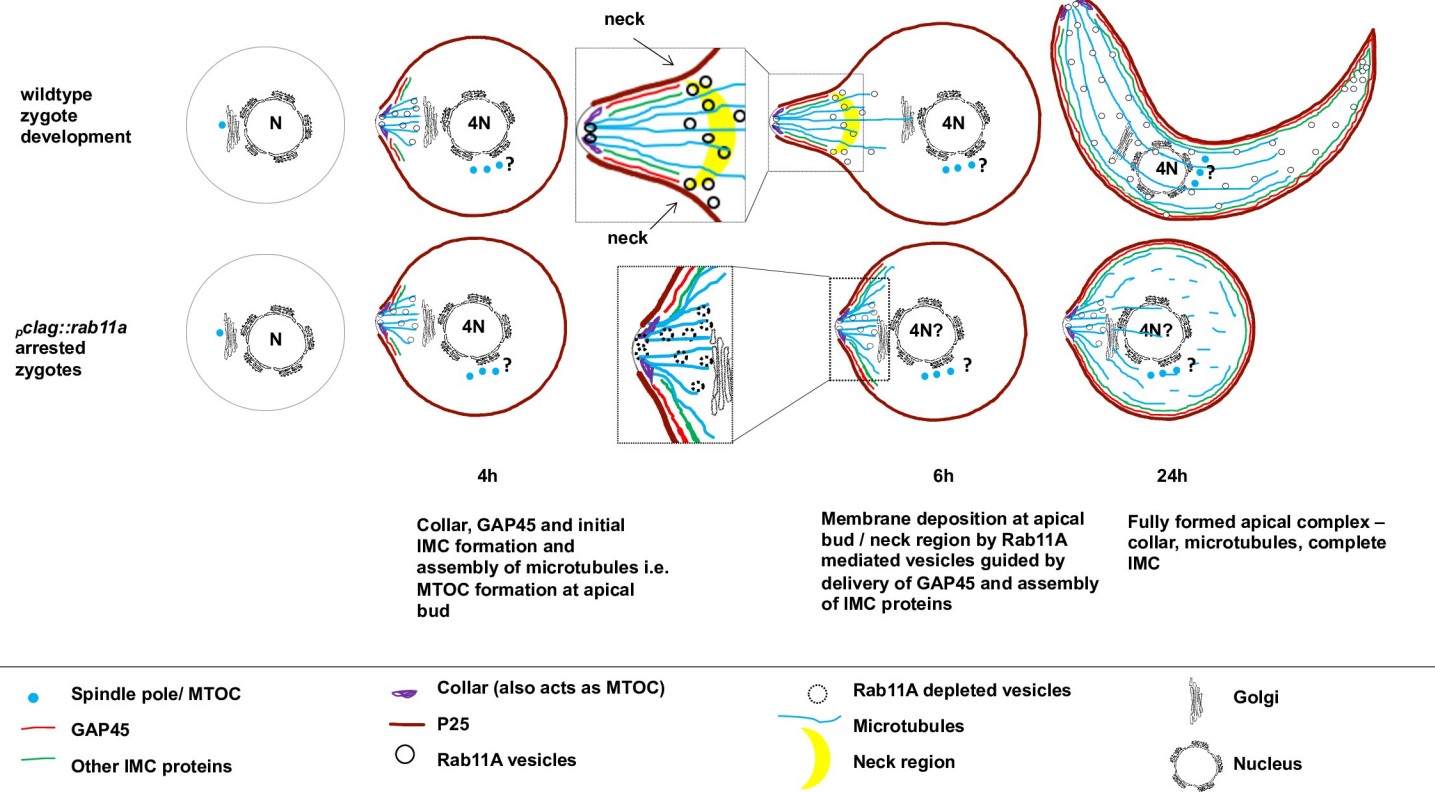

**Fig 7. Proposed Model: Rab11A mediated delivery of plasma membrane in *P. berghei*.** Rab11A mediated vesicles are involved in the delivery the plasma membrane and transmembrane proteins to the growing tip or to the neck region of retort / zygote. These are also possibly involved in secretion, development and maintenance of cell shape, protein trafficking coordinating with cytoskeletal components. Blue dots with '?' suggest possible numbers of spindle poles in a zygote. WT-GFP$_{CON}$ 6hpa retort showing possible site for membrane expansion either at the apical tip or neck region. Expanded cartoons of the apical regions of mutant and wild type parasites at 6hpa highlight the vesicles that lack RAB11A.

however, that there may well be stage and species-specific features to even these essential processes.

## Materials and methods

### Ethics statement

All animal work was approved by the University of Glasgow's Animal Welfare and Ethical Review Body and by the UK's Home Office (PPL 60/4443). The animal care and use protocol complied with the UK Animals (Scientific Procedures) Act 1986 as amended in 2012 and with European Directive 2010/63/EU on the Protection of Animals Used for Scientific Purposes.

### Animal procedures and parasite reference line

Parasites were maintained in Theiler's original (TO) or NIH Swiss outbred female mice, approximately weighing 25 g and > 6 weeks old.

### Generation of mutant parasites

All the genes, 5'UTRs and 3'UTRs amplifications were performed using either Expand High Fidelity PCR System (Roche) or KAPA HiFi PCR system (KAPA Biosciences) and diagnostic

PCRs were performed using Taq DNA polymerase (Invitrogen) as per manufacturer's instructions.

3'UTR of *the snare* (PBANKA_141880) was amplified from genomic (g)DNA of WT *P berghei* by using forward primer GU1620 and reverse primer GU1621 (S3 Table) and re-cloned into pG72 to generate $_p$*clag:2cmyc*::*rab11a* construct ('$_p$' stands for promoter). Construct $_p$*ama-1:2cmyc*::*rab11a* was obtained by replacing *clag* 5'UTR in $_p$*clag:2cmyc*::*rab11a* with PCR amplified 1.7 Kb 5'UTR of *ama-1* (PBANKA_091500) ORF from gDNA of WT *P. berghei* using forward primer GU1622 and reverse primer GU1623 (S3 Table). Plasmids $_p$*clag:2cmyc*:: *rab11a* and $_p$*ama-1:2cmyc*::*rab11a* were verified by restriction digestion as well as Sanger sequencing (Eurofins MWG biotech). Purified and *HindIII-NotI* (New England Biolabs) line-arised genomic constructs were obtained by gel extraction (QIAquick Gel Extraction Kit) and subsequent ethanol precipitation. >5μg of linearized genomic constructs were transfected into WT-GFP$_{CON}$ *P. berghei* [18] schizonts as described in [21] and [37] to express PbRab11A under the control of either *clag* (PBANKA_140060) or *ama-1* promoter respectively. (**Note:** $_p$*clag:2cmyc*::*rab11a* and $_p$*ama-1:2cmyc*::*rab11a* were referred as $_p$*clag*::*rab11a* and $_p$*ama-1*:: *rab11a* throughout respectively.)

## *Plasmodium berghei* infections, cultures and stage enrichment

Mice were infected by *intraperitoneal (i.p.)* injection either by fresh infected blood with PBS or by approximately 200 μl of blood suspension having parasitaemia 3–5% from thawed cryovials. Post-infection mice develop parasitaemia approximately 0.5–3% on day 3, and for some experiments, mice were given a single dose of 100 μl phenylhydrazine (12.5 mg/ml) (mice body-weight ratio 125 mg/kg) *i.p.* to enhance numbers of reticulocytes usually 48 h prior infection. Parasitaemia was monitored daily by Giemsa staining of thin smears of mouse tail blood [21] and parasites were observed under oil immersion (Invitrogen) 100x objective of light microscope (Primo Star- Zeiss). Images were captured through PAXcam5 camera using PAXcam software and processed using Fiji/ImageJ software (National Institute of Health, http://fiji.sc/ Fiji).

Schizonts were generated as described in [21]. For enrichment of gametocytes, post 72 h of infection mice were treated with sulfadiazine (30 mg/litre) in drinking water for 48h to kill asexual parasite stages. For ookinete cultures, cardiac blood containing gametocytes was placed into ookinete culture medium (RPMI1640 containing 25 mM HEPES, 5 mM hypoxanthine, 20% FCS, 10 mM sodium bicarbonate, 100 μM xanthurenic acid at pH 7.6) and incubated at 21°C for 24h. Schizonts, gametocytes and ookinete development was observed by Giemsa staining. Ookinete conversion rates were calculated either from Giemsa staining or immunofluorescence microscopy using Cyanine3-tagged anti-P25 antibody (see immunofluorescence assay methods) of ookinete culture. Ookinete conversion rates were calculated according to the following formula:

$$\% \text{ ookinete conversion} = \left( \frac{number\ of\ ookinetes\ analysed\ through\ morphology}{number\ of\ ookinetes + number\ of\ female\ or\ zygotes} \right)$$

To analyse activation centres cardiac blood was placed immediately into ookinete culture medium (1:30 blood to ookinete medium ratio) and 10 μl was placed on a haemocytometer at 21°C. After 15 minutes, activation centres defined as actively moving clumps of cells were counted using 10X objective on the light microscope (Primo Star- Zeiss). Fertility of mutant gametes was assessed through *in-vitro* cross fertilization with *P. berghei* parasites able to produce either only fertile male (P47 KO) [38] or female (P48/45 KO) gametes [39], respectively.

The male to female gametocyte ratio was calculated by examining under 100x oil immersion lens Giemsa stained slides made from tail blood taken from phenylhydrazine and sulfadiazine treated mice 72 hrs post-infection having similar parasitaemia. Five different fields with well separated cells were counted from each of the two Giemsa slides per experiment per parasite line (i.e. ten fields per experiment per parasite line). Six biological replicates were performed for $_p$clag::rab11a vs WT-GFP$_{CON}$ and four replicates for $_p$ama-1::rab11a vs WT-GFP$_{CON}$.

## Transmission

Transmission of parasites was performed by feeding 200 or 300 *Anopheles stephensi* mosquitoes on anaesthetized, infected mice having 2–8% parasitaemia. Mosquitoes were dissected on day 11 or 14 post-feeding for oocyst counting and on day 17, 18 and 22, 27 post-feeding for oocysts and sporozoite counting. Midguts and salivary glands were examined with Leica M205 FA Fluorescence Stereomicroscope and images were captured using Leica DFC340FX camera and through LAS AF Lite 2.2.0 build 4758 (Leica Microsystems Ltd.) and processed through ImageJ/Fiji software (National Institute of Health). Oocyst numbers were counted by live imaging.

## Motility assay

MACS-purified ookinetes were embedded in Matrigel and incubated for 1h at 21˚C before imaging. Time-lapse movies were acquired every 10 seconds for 10 minutes on a Leica M205 FA fluorescence stereomicroscope employing the GFP and mCherry filter sets (0.5 second exposure for each). Ookinete speeds were calculated on Fiji software (NIH, http://fiji.sc/Fiji) using the MtrackJ plugin [40].

## Scanning and transmission electron microscopy

For scanning and transmission electron microscopy *P. berghei* gametocytes 8 hours post-activation (hpa) and 24hpa in ookinete medium were processed as described in [41]. SEM samples were examined on a JEOL6400 SEM running at 10kV and tiff images captured using Olympus Scandium software. TEM samples were examined either on Tecnai T20 (FEI) running at 200kV or Leo 912AB TEM running 120kV and images captured using Gatan Digital Micrograph Software (Gatan, Japan).

## Inhibition of gametocyte fertilisation

To prevent fertilization of female gametes, unactivated gametocytes were cultured in enriched PBS (PBS supplemented with 2mM HEPES, 2mM Glucose, 0.4mM NaHCO$_3$, 0.01% BSA) containing 100 mM 2-Deoxy-D-glucose (Sigma) for 45 minutes [22] and filtered through Plasmodipur filters (EuroProxima), washed with ookinete medium and re-cultured in ookinete medium without 2-Deoxy-D-glucose (2DG) for 24h. Female gametocytes activate but fertilization is prevented due to a blockage in male gamete exflagellation [22].

## Western Blotting

For analysis of PbRAB11A and ookinete development markers, *P. berghei* schizonts were enriched with 55% Nycodenz solution in PBS; gametocytes and ookinetes were enriched by 53% Nycodenz solution in PBS following density gradient centrifugation [21]. Mixed asexual blood stage parasites collected from an infected mouse by cardiac puncture were filtered through CF11 cellulose columns (Whatman-GE Healthcare Life Sciences) to exclude mice

white blood cells and immediately lysed with ice-cold erythrocyte lysis buffer (1.5M $NH_4Cl$, 0.1M $KHCO_3$, 0.01 EDTA) for 15 minutes on ice. Equal amounts of parasite material from mixed asexual stage, schizonts, gametocytes and ookinetes were lysed in Net2+ buffer (140 mM NaCl, 50 mM Tris- pH 7.4, 4mM Dithiothreitol, 0.01% Nonidet P-40) supplemented with complete EDTA free Protease Inhibitor Cocktail Tablets (Roche) (1 tablet/10ml) and suspended in 2X Laemmli sample buffer containing 15% (v/v) β-mercaptoethanol and separated on 10 or 12% SDS polyacrylamide gels (Flowgen Biosciences) and visualized using ECL and X-ray films as described by [42]. Blots were probed with primary antibodies (S4 Table & S5 Table) and HRP coupled secondary antibodies either Polyclonal Goat anti-Rabbit or anti-mouse Ig/HRP (Dako) (1:10000). For infrared based visualization, appropriate secondary antibodies were used (IRDye 680LT Goat anti-mouse IgG and IRDye 800CW Goat anti-rabbit IgG antibodies) and nitrocellulose blots were scanned on Odyssey Sa Infrared Imaging System (LI-COR Biosciences). Blots were stripped twice for 5 minutes each with 0.2M NaOH solution and intermittent washing with distilled water and re-blocked in 5% milk in PBS-Tween before re-probing with further antibodies (S4 Table & S5 Table). Microneme secretion assays were performed as described by [41], parasite secretion supernatant samples were either concentrated on an Amicon-ultra centrifugal filter, 30 kDa cut-off (Millipore), suspended in Laemmli sample buffer and analysed as above or loaded and analysed directly.

## Immunofluorescence assay

Air dried thin smears of *P. berghei* ookinete or schizont or sporozoites and oocysts, obtained from midgut of *P. berghei* infected *Anopheles stephensi* crushed in PBS, were fixed with 4% EM grade paraformaldehyde (Electron Microscopy Science) for 10 minutes. Permeabilisation, blocking and incubation with primary and appropriate secondary antibodies (S4 Table & S5 Table) was performed as described by [41] with additional last washes with 70% ethanol and absolute ethanol 1 min each, air-dried and mounted in VectaShield (Vectorlabs) containing DAPI (4', 6-diamidino-2-phenylindole) in glycerol for nuclear staining. Parasites were examined either under Delta Vision Epifluorescence microscope (Applied Precision). 100x objective, images were captured with CoolSNAP HQ camera (Photometrics) and deconvoluted using SoftWoRx software (Applied Precision) or under Axioplane2 (Zeiss) 100x objective, images were taken through HAMAMATSU ORCA_ER camera (HAMAMATSU) and Velocity software 4.1.0 (PerkinElmer). Images were processed using Fiji (NIH) as well as SoftWoRx explorer 1.3. Super-resolution images were captured through Elyra PS.1 super-resolution microscope (Zeiss) under 60x objective with sCMOS PCO camera and images were visualized with ZEN Black software (Zeiss) and processed with ZEN LITE software (Zeiss).

## Flow Cytometry analysis

WT-GFP$_{CON}$, activated unfertilised WT-GFP$_{CON}$ female gametes (AUFG) and $_p$*clag::rab11a* gametes were collected 4hpa and 24hpa in ookinete media and purified using 53% Nycodenz density gradient centrifugation, washed with enriched PBS twice and vigorously vortexed three times 10 seconds each to break the zygote/ookinete clumps. Parasites were incubated with anti-P25 monoclonal antibody (mAb) (1:1000) diluted in enriched PBS for 30 minutes, washed and re-probed with secondary antibodies Goat anti-mouse Alexa Fluor 633 (1:1000) and erythroid cell marker anti-Ter119 PE/Cy5 antibody (eBioscience) (1:500) diluted in enriched PBS (see S5 Table for antibodies) supplemented with 5 μM Hoechst 33342 for 30 minutes, washed twice with enriched PBS and once with FACS buffer (2% (v/v) Fetal Bovine Serum (defined), 0.05% (w/v) sodium azide (NaN3) and 2 mM EDTA in PBS) vortexed. Parasites were re-suspended in 1ml FACS buffer and parasite aggregates were removed by filtering

through nylon filtration fabric NITEX 40 μm pore size (Cadisch Precision Meshes). Samples were run on a FACS CyAn (Beckman Coulter) equipped with a 405 nm, 488 nm and 633 nm laser and 5,000 to 10,000 events were acquired. Post-acquisition analysis was performed using Kaluza software (Beckman Coulter). The gating strategy was implemented to identify female gametes, zygote and ookinetes (GFP, P25 and Hoechst positive, Ter119 PE/Cy5 negative) from negative controls of uninfected RBCs (Ter119 PE/Cy5 positive), infected RBCs with asexual stages of parasites (Ter119 PE/Cy5 and Hoechst positive) and unstained samples. The DNA content was analysed by comparing the Hoechst 33342 stain levels of these parasites. Differentiation between mutant male and female gametocytes was not possible due to lack of two-colour background of mutants as shown by [43].

## Imaging flow Cytometry (IFC)

Ookinetes for IFC were prepared by harvesting ookinete cultures and disrupting cell clumps by passing through a 26g needle. Cells were stained with an anti-P25 antibody conjugated to the Cy3 fluorophore (αP25-Cy3) in enriched PBS for 45 minutes at room temperature. Parasites were washed in enriched PBS and resuspended in Magnesium and Calcium free Dulbecco's PBS (DPBS) and passed through 40 μm pore nitex (Cadisch Precision Meshes). Samples were run on an Imagestream[x] MkII (Merck) equipped with one camera and 4 lasers (405 nm, 488 nm, 561 nm and 633 nm). Single colour compensation controls were also acquired and post-acquisition analysis and gating was performed in IDEAS software (S4 Fig). Parasites stained for tubulin for IFC were harvested at indicated time points and incubated with 1:2000 dilution of Tubulin Tracker Deep Red (Invitrogen) and αP25-Cy3 in enriched PBS for 45 minutes at room temperature then resuspended in DPBS and filtered through nitex before acquisition on an Imagestream[X] MkII. 24 hour ookinete cultures were permeabilised by addition of L-α-Lysophosphatidylcholine at a final concentration of 250 μg/ml for 2 minutes in enriched PBS then washed in enriched PBS before staining as above.

## Transcriptomics

Unactivated gametocytes were immediately filtered through Plasmodipur filters at 37°C before enrichment step or used to set up ookinete cultures. All samples were enriched with Nycodenz density gradient centrifugation and mixed vigorously with 1 ml TRIzol (Ambion-Life Technologies) and stored at -80°C or immediately used for RNA isolation using RNAeasy Universal Mini kit (Qiagen) with on-column DNAse digestion by RNase-free DNase set (Qiagen) according to manufacturer's instructions. RNA quality was assessed through reverse transcriptase PCR performed using SuperScript III Reverse Transcriptase kit (Life Technology) before transcriptomic analysis (Glasgow Polyomics: www.polyomics.gla.ac.uk). RNA Sequencing (RNA-Seq) reads were prepared (Please see S6 Table and S7 Table for parameters of RNA-Seq) using Life Technologies stranded mRNA library kit and sequencing was carried out on Life Technologies Ion Proton platform.

Fastq files were quality controlled using FastQC (http://www.bioinformatics.babraham.ac.uk/projects/fastqc/, version 0.10.1) and trimmed for adapters and with a quality threshold >2 = 20 using cutadapt [[44], version 1.6 version, "-m 16 -b GGCCAAGGCG -q 20"]. Reads were then aligned to the *Plasmodium berghei* ANKA genome [[45] PlasmoDB version 11.1] using Tophat [[46]. Version 2.0.12, "–keep-fasta-order -b2-D 20 –b2-R 3 –b2-N 1 –b2-L 20 –b2-I S,1,0.50 -g 10 -I 5000–-library-type fr-firststran""]. In order to have maximum sensitivity to low abundance or divergent transcripts reads that failed to align with Tophat2 were extracted using bed tools (http://bedtools.readthedocs.org/en/latest/, version 2.19.1,""bamtofast"") and aligned to the same reference using bowtie2 local alignments [[47], version 2.2.1,

"–-local -D 20 -R 3 -N 1 -L 20 -i S,1,0.50–-m""]. The Tophat accepted hits and bowtie2 aligned reads were then merged to form the alignment for that sample using Picard tools (http:// broadinstitute.github.io/picard/, version 1.112). Gene-level expression analysis was carried out using the cufflinks2 package [[48] version 2.2.1] using annotated genes only (PlasmoDB VERSION 11.1 GFF file). Differential expression was carried out using cuffdiff (version 2.2.1). Subsequent analyses were carried out using the CummeRbund package (http://compbio.mit.edu/ cummeRbund/, version 2.0).

Generic GO slim terms were obtained through PlasmoDB.org and only Biological Process terms were shown for simplicity.

## Supporting information

**S1 Fig. Distribution of Rab11A in wild type ookinete.** Localization of Rab11A in WT-GFP$_{CON}$ ookinete. The first column shows magnified images of apical tip of ookinete (taken from the images in the second column) stained with anti-PbRab11A antibody. Scale bar 5 μm. (TIF)

**S2 Fig. Genomic introduction of *pbrab11a* promoter swap constructs. A.** *P. berghei* RNA-Seq data for *rab11a*, *clag* and *ama-1* shown in FPKM [49]. **B.** Schematic of the generation of $_p$*clag*::*rab11a* and $_p$*ama-1*::*rab11a* parasites (Not to Scale). Diagnostic PCRs for integration of **(C)** $_p$*clag*::*rab11a* and **(D)** $_p$*ama-1*:: *rab11a* constructs into WT-GFP$_{CON}$ gDNA, respectively, showing the 5' and 3' integration of respective constructs (PCR fragment as annotated in the schematic in **B**). W indicates a fragment present only in WT-GFP$_{CON}$ parasites. DNA '+' is an unrelated positive PCR control (P28) and '-' is a no DNA template negative PCR control. Abbreviations: USR, upstream region; SM, selectable marker. (TIF)

**S3 Fig. Development of sexual stages of $_p$*clag*::*rab11a* and $_p$*ama-1*::*rab11a* parasites mirrors that of wild type parasites. A.** Images of Giemsa stained $_p$*clag*::*rab11a* and $_p$*ama-1*::*rab11a* gametocytes; M, male gametocytes; F, female gametocytes. **B.** Exflagellation count of male gametocytes of $_p$*clag*::*rab11a* (n = 4, mean +/-SD, two tailed student t test, p-value 0.9891) and $_p$*ama-1*::*rab11a* (n = 3, mean +/-SD, two tailed student t test, p-value 0.4337). **C.** Ratio of male to female gametocytes in $_p$*clag*::*rab11a* (n = 6, mean +/-SD, two tailed student t test, p value 0.00046) and $_p$*ama-1*::*rab11a* (n = 3, mean +/-SD, two tailed student test, p-value 0.0111) parasites compared to WT-GFP$_{CON}$. (TIF)

**S4 Fig. Expression of PbRAB11B is unaffected by downregulation of PbRAB11A in $_p$*clag*:: *rab11a* mutant parasites. A.** Western blot analysis of PbRAB11B expression following activation of gametocytes comparing wild type parasites with $_p$*clag*::*rab11a* and Activated Unfertilised Female Gametocytes (AUFG) across the time course of zygote to ookinete transition sampling at times indicated in hours. **B.** Immunofluorescence imaging of PbRAB11B expression at 24hpa in the same parasite lines when mature ookinetes should be present. Scale bar = 5μm. (TIF)

**S5 Fig. Imaging flow cytometry gating strategy to quantitate ookinete conversion rate.** Dot plots and histograms showing the sequential gating strategy pipeline used to identify ookinetes and quantitate conversion rate. Files were acquired separately then merged in IDEAS software. Step 1 –gate R1 defined objects within a broad size range that includes infected RBC, gametocytes and ookinetes. Step 2- R2 includes cells positive for GFP. Step 3 –R3 gates on cells in

focus on the Brightfield image. This gate is broad as we observe that some activated female gametocytes displayed a lower gradient RMS that is normal for focussed cells. Step 4—Gate R4 allows us to select cells in focus in the GFP channel. Step 5—Gate R5 selects for cells positive for P25-Cy3. The proportion of GFP positive cells that were also positive for P25-Cy3 was similar for the WT and the promoter swap lines. Step 6—The next gate selects cells also in focus in the P25-CY3 channel. GFP positive cells include asexual stages and non-activated gametocytes. Events positive for GFP and Cy3 not included for analysis include uninfected RBC that are autofluorescent as a result of the phenylhydrazine treatment given to enhance parasitaemia. Step 7—To help exclude images containing debris or overlapping cells a mask was generated to allow selection of cells where the area of the brightfield image was sufficiently larger than that of the area of the GFP image (Area_adaptiveErode(M05,Ch05) And not Area (M02, Ch02)). (Other strategies to exclude debris included using a spot count feature to identify images containing a single object, or the threshold feature to analyse only the object within the image that was within the size and intensity criteria. In some cases, images including doublets and debris were manually selected for exclusion from analysis). Step 8- To separate out the three merged files the object number vs time is plotted and three populations from the individual samples can be separated. Step 9 –Finally the circularity and aspect ratio features were generated on the adaptive erode (84%) mask for the GFP image and used to plot ookinete conversion. Similar strategies using the brightfield image and the Cy3 image were also successful and gave similar results. All gating strategies were justified by examining images of objects falling outwith the gates. Step 10 –In order to plot the ookinete development against other features the aspect ratio and circularity features were combined onto one axis. Statistics below show the number of cells included and the % of each population falling into each gate. In this figure 507 refers to WT-GFP$_{CON}$, G480 refers to $_p$*clag*::*rab11a* and G481 refers to $_p$*ama1*::*rab11a*.
(TIF)

**S6 Fig. IFC Images.** Gallery of representative images from the ookinete gate IV (Fig 2G) for the WT parasite line (top) and the promoter swap mutant lines. Scale bar (lower left) 7 μm.
(TIF)

**S7 Fig. $_p$*clag*::*rab11a* spherical ookinetes undergo meiosis. A.** Flow cytometry analysis on FACsCYAN to illustrate DNA content of $_p$*clag*::*rab11a* gametocytes 4hpa. Parasites stained with DNA stain Hoechst 33342 were gated for activation using anti-P25 antibody and DNA content in these compared to WT-GFP$_{CON}$ gametocytes 4hpa and WT-GFP$_{CON}$ activated unfertilized female gametes (AUFG) 4hpa. FACS plots showing results of one of three independent experiments. **B.** bar graph shows percentage of 4N (zygotes completed meiosis), 2N (fertilized female gametes, meiosis is incomplete or blocked) and 1N (gametocytes or asexual) parasites. Data from AUFG was used to verify the gating strategy (n = 3, mean +/-SD, two tailed student t test, p-value 0.129292).
(TIF)

**S8 Fig. Active *rab11a* is contributed by both male and female gametes. A.** Cross-fertilization of $_p$*clag*::*rab11a* with male defective (*P48/45$^-$*) and female defective (*p47$^-$*) mutants (n = 3, mean +/- SD, two tailed student t test, p value 0.0001) and **B.** Representative Giemsa images of ookinetes and zygotes obtained 24h after (cross) fertilisation. The decondensed nuclei of the $_p$*clag*::*rab11a* zygotes are arrowed. Scale bar = 3μm.
(TIF)

**S9 Fig. $_p$*ama-1*::*rab11a* parasites are unable to transmit through mosquitoes. A.** Plot of oocyst load in dissected midguts of WT-GFP$_{CON}$ and $_p$*ama-1*::*rab11a* fed mosquitoes (n = 2, two tailed student t test, p-value 0.0001). **B.** Fluorescent and bright field images of

WT-GFP$_{CON}$ and $_p$*ama-1::rab11a* infected midguts and salivary glands.
(TIF)

**S10 Fig. Immunofluorescence microscopy for ookinete development and structural markers.** Fixed WT-GFP$_{CON}$ ookinetes and $_p$*clag::rab11a* spherical ookinetes were probed with primary antibodies: anti-DOZI, anti-CITH, anti-PPKL, anti-CTRP, anti-MyoA and anti-MTIP antibodies mixed with either FITC-tagged anti-P25 or anti-α tubulin antibodies. Except PPKL, all images shown are single slice of Deltavision deconvoluted Z stack. For PPKL, single slice images of Z stacks obtained from ELYRA 3D SIM microscope is shown. Scale bar 5 = μm.
(TIF)

**S11 Fig. Dynamic localization of GAP45.** Time course immunofluorescence of WT-GFP$_{CON}$ and $_p$*clag::rab11a* zygotes for GAP45 and P25 [4hpa timepoint images are taken through Axioplan and 6hpa timepoint images are single slices of deconvoluted Z stack taken from Deltavision microscope]. Scale Bar = 5 μm.
(TIF)

**S12 Fig. Significantly deregulated genes in WT gametocytes vs $_p$*clag::rab11a* gametocytes.** Summary of the expression levels of genes that are more than 2 fold altered in a comparison of wild type and *rab11a* KD gametocytes as assessed by RNAseq analysis.
(TIF)

**S13 Fig. The altered transcription profile of *pbrab11a* KD gametocytes does not significantly overlap with that of either DOZI and CITH KO gametocytes.** Venn diagram representation of the extent of similarity between the transcriptomes of $_p$*clag::rab11a*, *cithko* and *doziko* gametocytes.
(TIF)

**S14 Fig. Significantly deregulated genes in WT ookinetes vs $_p$*clag::rab11a* 24hpa forms.** Summary of the expression levels of genes that are more than 2 fold altered in a comparison of wild type and *rab11a* KD ookinetes as assessed by RNAseq analysis.
(TIF)

**S1 Table. Showing $_p$*clag::rab11a* parasites are unable to transmit through mosquitos.**
(XLSX)

**S2 Table. Showing $_p$*ama1::rab11a* parasites are unable to transmit through mosquitos.**
(XLSX)

**S3 Table. Primers used for PCR amplifications, generation of plasmids and diagnostic PCRs.**
(XLSX)

**S4 Table. Peptide raised primary antibodies from Proteintech used for immunofluorescence or western blotting.**
(XLSX)

**S5 Table. Primary antibodies obtained from external sources, their dilutions used for immunofluorescence, western blotting or FACS analysis.**
(XLSX)

**S6 Table. RNASeq reads trimmed and raw for three sets of parasite samples.**
(XLSX)

**S7 Table. Number of reads aligned to the genome of P. berghei ANKA for comparison with the original read counts.**
(XLSX)

## Acknowledgments

The authors gratefully acknowledge the Glasgow Imaging Facility and the Flow Core Facility for their support and assistance in this work. HP was an Evimalar Ph.D. student and the generous input of Prof. Freddy Frischknecht is gratefully acknowledged. Michael Rennie prepared some of the ookinetes for EM imaging.

## Author Contributions

**Conceptualization:** Katie R. Hughes, Andrew. P. Waters.

**Formal analysis:** Nicholas Dickens.

**Investigation:** Harshal Patil, Katie R. Hughes, Leandro Lemgruber, Nisha Philip.

**Methodology:** Katie R. Hughes, Nisha Philip.

**Project administration:** Andrew. P. Waters.

**Resources:** G. Lucas Starnes.

**Supervision:** Katie R. Hughes, Andrew. P. Waters.

**Validation:** Katie R. Hughes.

**Writing – original draft:** Harshal Patil, Andrew. P. Waters.

**Writing – review & editing:** Katie R. Hughes, Nicholas Dickens, Andrew. P. Waters.

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
