## [Decision Letter · Decision Letter 0]

4 Nov 2019

Dear Andy

Thank you very much for submitting your manuscript "Zygote morphogenesis but not the establishment of cell polarity in Plasmodium berghei is controlled by the small GTPase, RAB11A" (PPATHOGENS-D-19-01709) for review by PLOS Pathogens. Your manuscript was fully evaluated at the editorial level and by independent peer reviewers. The reviewers appreciated the attention to an important topic but identified some aspects of the manuscript that should be improved. In partcular, the reviewers queried the accuracy of much of the detail of the text throughout, as well as the clarity of the image presented in Fig 1D. Quantitation of the data shown in Fig 6 as well as an indication of the reproducibility of those results, is also important. We do not agree that universal use of the past tense in the Results section is necessary, but please attend to the proof-reading requests made.

We therefore ask you to modify the manuscript according to the review recommendations before we can consider your manuscript for acceptance. Your revisions should address the specific points made by each reviewer.

(1) A letter containing a detailed list of your responses to the review comments and a description of the changes you have made in the manuscript. Please note while forming your response, if your article is accepted, you may have the opportunity to make the peer review history publicly available. The record will include editor decision letters (with reviews) and your responses to reviewer comments. If eligible, we will contact you to opt in or out.

(2) Two versions of the manuscript: one with either highlights or tracked changes denoting where the text has been changed; the other a clean version (uploaded as the manuscript file).

We hope to receive your revised manuscript within 60 days or less. If you anticipate any delay in its return, we ask that you let us know the expected resubmission date by replying to this email.

[LINK]

Sincerely,

Michael J Blackman

Associate Editor

PLOS Pathogens

Kami Kim

Section Editor

PLOS Pathogens

Kasturi Haldar

Editor-in-Chief

PLOS Pathogens

orcid.org/0000-0001-5065-158X

Grant McFadden

Editor-in-Chief

PLOS Pathogens

orcid.org/0000-0002-2556-3526

Reviewer's Responses to Questions

**Part I - Summary**

Reviewer #1: This manuscript describes a Plasmodium berghei knock-down of the gene encoding Rab11A. While this gene has previously been studied in Plasmodium the study offer new and unexpected insights into the function of the protein. The authors used a previously developed promoter swap approach to impair expression of the protein in the gametocyte stage and further onwards to the ookinete. This offered the possibility to study the function of the protein as a knock-out of the gene has been shown to be lethal. Two different promoters were used to generate the mutant parasites, with both showing the same phenotype. The mutants were analyzed in detail and the results show that they are blocked in the development from the round zygote to the extended ookinete. They show convincingly that polarity of the cell, defined as the presence of the apical complex, is not affected in the absence of the protein. The mutant parasites lack motility and secretion of micronemal proteins is also affected. The authors suggest that Rab11A is necessary for the trafficking of vesicles for the extension of the plasma membrane during ookinete development which is a reasonable hypothesis based on the data.

The study is of high quality and the conclusions are based on solid experimental evidence. Rab11A and its function in apicomplean parasites is of general interest. It is also of interest that I reviewed a study of the same protein in T. gondii and the publication of the two papers will provide a plethora of new data on this important protein in these parasites. I therefore recommend that the paper is accepted.

The authors also introduce a new method to study these cells using Imaging flow cytometry to obtain quantitative data.

Reviewer #2: The paper from Patil et al. Investigates the role of the small GTPase PbRab11A in Plasmodium berghei gametocytes and ookinetes using a promoter swap strategy to downregulate expression of PbRab11A in the targeted stages. Lack of PbRab11A had no effect on the production of gametes nor the fertility rate. Interestingly, lack of PbRab11A inhibited transmission of the parasite. This was correlated to a default in the zygote to ookinete transition, impaired motility and microneme secretion, and lack of morphological maturation.

This study provides valuable data regarding zygote and ookinete formation in Plasmodium berghei. It uses an elegant technological approach, i.e. promoter swap with promoters known to be silent in sexual stages. The subsequent experiments performed to investigate the consequences of knocking down Pbrab11A are well conducted and dissect all the crucial steps of the ookinete formation, with clear results.

Reviewer #3: The authors present an extensive characterisation of morphological developmental changes in ookinetes with Rab11A depletion. This builds on knowledge that the protein is unable to be knocked out in blood-stage parasites and uses molecular tools that enable propagation of a Rab11A positive cell line in blood stages that then losses Rab11A expression during the sextual stages. The major observations are that viable ookinete development is essential blocked, and cells fail to differentiate into the elongated ookinete form. This coincides with loss of motility, but interestingly much of the ultrastructure of the apical complex appears to be formed and intact. This is somewhat surprising given that elements of the IMC were presumed to rely on Rab11A for delivery and synthesis. Some evidence of a secretion defect is also presented, although these data are quite preliminary and undeveloped.

Overall, the study helps to define some elements of ookinete development that are not reliant on Rab11A (polarity establishment, apical complex assembly), and some that are (elongation and likely secretion competence for some cargo at least). A hypothesis is suggested that Rab11A is required for delivery of lipid for membrane expansion, and absence of this contributes to failure to elongate. This hypothesis is plausible but awaits testing. The data that suggests that secretion is affected by Rab11A depletion offers a further line of enquiry to resolve the roles of this protein, but this is represented by only one experiment, and the conclusions presented here are not well supported by the data at this stage (see below). This seems like an obvious and underexplored facet of Rab11A’s function in parasites, and perhaps one that is mechanistically linked with lipid delivery to the plasma membrane also.

The study has been executed very well, with some minor exceptions detailed below. It provides useful insight into elements of the process of cell formation, although ultimately a clear understanding of the role of Rab11A is outstanding. Nevertheless, this study will doubtlessly contribute to resolving this question. I would have liked to see a little more exploration of how the observations made in ookinete development might inform on the role and requirement of Rab11A in the blood-stage, and think that this would offer some further impact to this study.

**Part II – Major Issues: Key Experiments Required for Acceptance**

Reviewer #1: None

Reviewer #2: no major issue

Reviewer #3: Some of the conclusions presented in the text were not well supported by the data.

A focus of Rab11A at the apical tip of ookinetes is described in the text but is not evident from Fig 1D. This looks like just the convergence of the periphery of the cell here.

The conclusion that a full and functional apical complex being developed in the mutant is going beyond what the data can report. Rather, by TEM the features of the apical complex can't be distinguished between the wildtype and mutants, leading to the hypothesis that it is full and functional. But at this point this hypothesis is untested.

The quantitation of protein amounts shown in Westerns in Fig 6 is not compelling. Stated reductions in CTRP are not evident comparted to DOZI. Only one experiment is shown. For this element of the study to be robust some further, independent data is required, and some numeric presentation would be ideal.

**Part III – Minor Issues: Editorial and Data Presentation Modifications**

Reviewer #1: Page 4. The sentence “The role of RAB11A in apicomplexan parasites and Plasmodium falciparum rab11a (PF3D7_1320600) is essential…” needs revision

Page 4. The protein MTIP is defined as Microtubule Interacting Protein while the correct definition is Myosin A Tail domain Interacting Protein.

Page 5. “Rab11A controls the necessary step after biogenesis…” add a reference.

Page 7. “The nucleus of pclag::rab11a “ookinetes” appears decondensed”. This is not obvious in Fig. S7 where Giemsa stained cells are shown. Possibly Hoechst label would be more informative of this aspect. Either remove the sentence or show a better picture.

Page 8. The speed of WT ookinetes was measured and shown in Fig. 3C. In the text it says the average speed was 10 um/min while in the figure it is estimated at ~4 um/min.

Page 11. “RAB11A is a highly conserved protein… host cell surface” it is not clear which host cell the authors refer to.

Page 12. “Spindle microtubules” are mentioned in several places referring to the microtubules of the ookinete. In my opinion this is not a correct term as spindle microtubules usually refer to the microtubules involved in mitosis and meiosis.

Page 16. “Fertility of…” this sentence needs revision.

Fig. 1D. There is a non-labeled sporozoite for Rab11A in this Figure which shows protein expression during the life cycle by IFA. It is somewhat unexpected that the protein will not be expressed in sporozoites so there should be a comment in the text about this finding. The asterisk in the same picture is not defined.

Figure 5F legend. Reverse the order of the sentence so that “upper” is first.

Fig. 5F. In the non-permeabilized samples P25 is found in a spot in the WT parasites, while the protein is found in the periphery of the whole cell in the mutant. In fact the latter pattern should reflect the normal distribution of P25 in the ookinete. Could this difference suggest that the protein is associated to the membrane more “loosely” in the WT ookinete? Please comment.

Fig. S2C. The comparison of the ratios of male/female looks almost identical in the graph but the P values are very low. What exactly is compared here?

Supplementary video 1. It is very difficult to see the moving ookinete(s). As the video is from a field with many cells it would be preferable if an arrow indicates the moving ookinete. Or show at higher resolution just the cell of interest. In fact the video does not add any information and could be removed.

Reviewer #2: 1/ The paper needs serious proofreading in order to ease its understanding, in particular (but not only) in the results section. Here are a couple of problems that need to be corrected:

- Fig 1D: why do you show a midgut sporozoite, which seems to not express PbRab11A (black)? If this is relevant, say it and discuss it.

- Fig. S1B: I believe, the general assumption for the abbreviation “UTR” is “UnTranslated Region”. The 5’ UTR thus corresponds to the region, which is comprised between the transcription start site and the ATG start codon of a gene, and thus excludes the promoter. The scheme as well as the text must thus be corrected so that there will not be any confusion possible on which part of the gene has been exchanged. What does mean SM?

- Page 5, you mentioned in the results section that you chose the pclag::rab11A line for further analysis. However, the results of the pama-1::rab11A line are often included in the experiments (which does not hurt, of course), just modulate the sentence stating that you will only study the pclag::rab11A line.

- Fig. 2D-E: mention at least that the anti-c-myc and anti-Rab11A detect a lower band in pclag::rab11A, not in pama-1::rab11A. How do you explain this difference?

- Fig. S3, as I did understand it, refers to the expression of the PbRab11B paralog. The results section do not describe these results.

- Fig. 2G: what about the stage IV? No expression of P25 (graph on the right).

- Fig. 2I: I do not see it mentioned in the text.

- Fig. S6: although the pclag::rab11A line seems to be able to produce 4N parasites indeed, the parasites seem to produce less 4N cells (and conversely, more 2N and 1N cells). The sentence in the results section should be modulated.

- Not being familiar with the proteins expressed by the sexual stages, I have a naïve question: you mentioned that the p47 KO line is female defective. I would thus expect it to produce only male gametocytes. However, I see a dark grey bar in the p47- column on Fig. S7A. I am confused.

- Fig.3A and Fig. S8A: at which day were these oocysts numbered? What does represent the horizontal bar, the mean or the median? In the results, please refer to Fig. S8A, when appropriate.

- Fig. 3D: how do you explain that so few WT-mCherry did move? Did you quantify these movements?

- Fig. 4B: you should label the WT-GPF photos with IMC, Mt, M, etc, so that the comparison with the pclag::rab11A line is easier.

- Fig. 5A: GAP45, which is indicated to be 23.6 kDa (or Kd, but not kD) seems to migrate at 45-50 kDa… please check also Fig. 2D and Fig. 2E for the kDa.

- Fig. 5F: a few scale bars are missing. To make the figure clearer, you should also indicate on the figure that the upper panel corresponds to “no permeabilization” and the lower panel to “partial permeabilization”.

- Fig. 6: Quantification of the signals would allow to normalize on the DOZI signal resulting from lysis. Please add the “kDa” on the left.

- The model should be included as Figure 7. The Golgi apparatus “G” is not indicated.

- Fig. S4: indicate the step numbers, as mentioned in the text for better clarity.

- Discussion, end of page 12: “SEM demonstrated a regular array of microtubule origins around the apical prominence.” Confusion with the TEM results? (as SEM cannot show internal structures)

- Discussion, end of page 13: if I understand well, you hypothesize that this is the lack of fusion of PbRab11A endosomes with the plasma membrane, which would be responsible for the lack of outgrowth of the apical bud, thus producing these roundish ookinetes. However, elongation of the cell would require a lot more lipids, i.e. its own lipid production, the endosome membranes are likely not sufficient to elongate the cell. Modification of both the lipid composition and the interactions with the cytoskeleton are also expected. This part of the discussion should be modulated. These are intriguing aspects that could be further investigated.

Question/suggestion out of the reviewing: would that be possible to develop super-resolution microscopy for this parasite? If yes, it would help a lot investigate the development of extracellular stages.

2/ please perform a complete proofreading of the manuscript to 1) write the results in the past tense and 2) to correct the grammatical, orthography, punctuation problems (they are more numerous in the results section).

Reviewer #3: The results tend to be narrated in terms of what experiments/methods were used, and what data resulted, without a clear narrative of the questions that were driving the experiments. This made it a bit more cumbersome as a reader to decipher what was been tested and why.

Page 4 "during recycling endosomes" seems grammatically incomplete.

Fig 2F, I am not readily learning what the colours of the dots represent from this figure legend.

Page 12: "Spindle microtubules" should be clearly defined to avoid confusion with mitotic spindles.

The grey vesus white rendering of the cytoplasm in Fig 7 is a little unclear.

PLOS authors have the option to publish the peer review history of their article (what does this mean?). If published, this will include your full peer review and any attached files.

Reviewer #1: No

Reviewer #2: No

Reviewer #3: No

---

## [Decision Letter · Decision Letter 1]

11 Feb 2020

Dear Andy,

Thank you very much for submitting your manuscript "Zygote morphogenesis but not the establishment of cell polarity in Plasmodium berghei is controlled by the small GTPase, RAB11A" for consideration at PLOS Pathogens. As with all papers reviewed by the journal, your manuscript was reviewed by members of the editorial board and by an independent reviewer. In light of the reviews (below this email), we would like to invite the resubmission of a revised version that takes into account the reviewers' comments.

As you will see, despite the previous round of revision, the reviewer raises questions about the data presented in Figure 6, which do not appear concordant with the main text. We agree that the figure does not present any data on CTRP and that some of the control data also appear inconsistent with the conclusions of the manuscript. We would be grateful if you could attend to these concerns. Please also consider the invitation to add additional ookinete images to the supplemental data (with regard to Figure 1D).

We cannot make any decision about publication until we have seen the revised manuscript and your response to the reviewers' comments. Your revised manuscript is also likely to be sent to reviewers for further evaluation.

Sincerely,

Michael J Blackman

Associate Editor

PLOS Pathogens

Kami Kim

Section Editor

PLOS Pathogens

Kasturi Haldar

Editor-in-Chief

PLOS Pathogens

orcid.org/0000-0001-5065-158X

Michael Malim

Editor-in-Chief

PLOS Pathogens

orcid.org/0000-0002-7699-2064

Reviewer's Responses to Questions

**Part I - Summary**

Reviewer #3: Generally most comments have been appropriately revised. But with the exception of the data presented in Fig 6.

**Part II – Major Issues: Key Experiments Required for Acceptance**

Reviewer #3: Regarding Figure 6 and the reduction of expressed and secreted CTRP and chitinase, I’m afraid that I am even more lost than before (actually I understood the first version, but not this one). The authors state in the Results “Using CTRP and chitinase as the markers for the two fates of apical organelle cargo . . . Reduced production of both chitinase and CTRP was evident in both mutants as well as significantly reduced chitinase secretion (Fig 6).” and in the Discussion “”the expression of microneme proteins CTRP (also associated with ookinete motility) and chitinase appears to be reduced in pclag::rab11a and pama1::rab11a ookinetes than WT-GFPCON ookinetes . . . Secretion assays confirmed the reduced expression of CTRP and chitinase and demonstrated that there was no obvious secretion of chitinase“. However, CTRP data is not included in the figure. The rebuttal states that “We reproduce the original western here but have made new samples and for simplicity run the assay omitting the CTRP detection”, but the text implies that the argument is still based on CTRP data, and therefore this should be included. I’m not particularly in favour of choosing ‘simplicity’ over an independent test of a hypothesis which CTRP appears to provide. But beyond these above queries, I don’t understand the data now presented in Figure 6. My understanding is that DOZI serves as a lysis control, but significant amounts of DOZI are in the supernatants for several samples implying significant lysis? Relatives amounts of protein expression between cell types is derived from these data, but there is no loading control stated that allows these relative comparisons. Or is DOZI in the pellet serving this purpose? But if so, variable lysis between treatments evident suggest this is not appropriate or reliable. Finally, the value of showing two independent experiments is to test the reproducibility of the experiment. But there are seemingly stark differences between the quantitation of these two experiments, but without analysis or discussion of these differences. For e.g. in B it looks like relative secretion of chitinase in the mutants, relative to their pellet material, is high (contrary to the conclusions), but this outcome is different in A.

Apologies to the authors if I am missing something obvious in these data or their presentation, and the Editor is free to override these comments if so. Otherwise, I do think that this element of the report is still inappropriate to be presented in its current form because conclusions made do not seem to be supported by the data presented.

**Part III – Minor Issues: Editorial and Data Presentation Modifications**

Reviewer #3: Regarding Figure 1D, if the authors are willing to include extra images of the ookinete in the supplemental data, I would advocate this. It is a subtle staining result, and is more convincing for the multiple images.

PLOS authors have the option to publish the peer review history of their article (what does this mean?). If published, this will include your full peer review and any attached files.

Reviewer #3: No
---

## [Editor Report · Decision Letter 2]

29 Feb 2020

Dear Andy,

We are pleased to inform you that your manuscript 'Zygote morphogenesis but not the establishment of cell polarity in Plasmodium berghei is controlled by the small GTPase, RAB11A' has been provisionally accepted for publication in PLOS Pathogens.

Best regards,

Michael J Blackman

Associate Editor

PLOS Pathogens

Kami Kim

Section Editor

PLOS Pathogens

Kasturi Haldar

Editor-in-Chief

PLOS Pathogens

orcid.org/0000-0001-5065-158X

Michael Malim

Editor-in-Chief

PLOS Pathogens

orcid.org/0000-0002-7699-2064
---

## [Editor Report · Acceptance letter]

29 Apr 2020

Dear Professor Waters,

We are delighted to inform you that your manuscript, "Zygote morphogenesis but not the establishment of cell polarity in Plasmodium berghei is controlled by the small GTPase, RAB11A," has been formally accepted for publication in PLOS Pathogens.

Best regards,

Kasturi Haldar

Editor-in-Chief

PLOS Pathogens

orcid.org/0000-0001-5065-158X

Michael Malim

Editor-in-Chief

PLOS Pathogens

orcid.org/0000-0002-7699-2064